# Balanced k-Means Revisited

## Abstract

The $k$-means algorithm aims at minimizing the variance within clusters without considering the balance of cluster sizes. Balanced $k$-means defines the partition as a pairing problem which enforces the cluster sizes to be strictly balanced, but the resulting algorithm is impractically slow $\mathcal{O}(n^3)$. Regularized $k$-means addresses the problem by using a regularization term including a balance parameter. It works reasonably well when the balance of the cluster sizes is a mandatory requirement but does not generalize well for soft balance requirements. In this paper, we revisit the $k$-means algorithm as a two-objective optimization problem with two goals contradicting each other: to minimize the variance within clusters and to minimize the difference in cluster sizes. The proposed algorithm implements a balance-driven variant of $k$-means which initially only focuses on minimizing the variance but adds more weight to the balance constraint in each iteration. The resulting balance degree is not determined by a control parameter that has to be tuned, but by the point of termination which can be precisely specified by a balance criterion.

## 1 Introduction

Clustering denotes the unsupervised classification of objects into groups, called clusters. Objects can be, for example, observations, data items, feature vectors or images (Jain et al., 1999). They are classified based on their similarity: objects within one cluster are more similar to each other than objects in different clusters (Kovács et al., 2006). Clustering is an important topic in several areas such as statistics, pattern recognition, machine learning and data mining (Kovács et al., 2006). Applications can be found, for example, in document retrieval, object and character recognition and image segmentation (Althoff, 2010; Jain et al., 1999).

Since the middle of the last century, thousands of algorithms facing the clustering problem have been published (Jain, 2010). They vary, among other things, in the way how they define the clusters or how they measure the similarity between objects (Aggarwal, 2015). For example, hierarchical clustering leads to a nested series of partitions, partitional clustering produces only one partition by decomposing the data set directly into a set of disjoint clusters and fuzzy clustering uses soft assingments (Gan et al., 2007; Jain et al., 1999; Kovács et al., 2006). Moreover, different approaches result in different types of clusters. Probabilistic model-based algorithms rely on the assumption that the data was generated from a mixture of distributions. Grid-based and density-based algorithms are able to detect fine-grained dense regions in the data. Graph-based algorithms are applicable to almost every type of data by converting the data set into a similarity graph. Representative-based algorithms rely on representatives for each cluster, which can be created by a function of the objects belonging to that cluster or can be objects themselves. Depending on the kind of the algorithm, the result of a clustering algorithm on the same data set can vary considerably. (Aggarwal, 2015; Kovács et al., 2006)

One of the most popular clustering algorithms is the $k$-means algorithm (Jain, 2010; Wu et al., 2007). It was first proposed by Lloyd (1982) and Forgy (1965). The aim of this clustering algorithm is to build $k$ disjoint clusters such that the sum of squared distances between the data points and their representatives is minimized. The representatives, called centroids, are determined by the mean of the data points belonging to a cluster. As a distance function the Euclidean distance is used. The number of clusters $k$ has to be set by the user.

Formally, if we have a data set consisting of $n$ data points $x_1, x_2, \ldots, x_n$, the task is to group these data points into $k$ clusters such that the sum squared error (SSE)

$$\text{SSE} = \sum_{j=1}^{k} \sum_{x_i \in p_j} ||x_i - c_j||^2 \tag{1}$$

is minimized, where $p_j$ denotes the set of data points assigned to the $j^{\text{th}}$ cluster and $c_j$ is the centroid of the $j^{\text{th}}$ cluster. This problem is NP-hard even for two clusters (Costa et al., 2017). The $k$-means algorithm is a heuristic for this problem.

The algorithm itself starts by a first initialization of the clusters, followed by an assignment and update step, which are iteratively repeated until a convergence criterion is met. During the initialization, $k$ cluster centroids are randomly selected from all data points. In the assignment step, each data point is assigned to the cluster whose centroid is closest. Formally, the assignment step can be written as

$$p_j = \left\{ x_i \mid \underset{j^* \in \{1, \ldots, k\}}{\arg\min} \left( ||x_i - c_{j^*}||^2 \right) = j \right\} \text{ for all } j \in \{1, \ldots, k\}.$$

In the update step, the centroids are updated by the mean of the data points assigned to the cluster. Formally,

$$c_j = \frac{1}{|p_j|} \sum_{x_i \in p_j} x_i \text{ for all } j \in \{1, \ldots, k\},$$

where $|p_j|$ denotes the number of data points assigned to the $j^{\text{th}}$ cluster. The assignment and update steps are repeated until the centroids do not change anymore.

This algorithm is a heuristic not necessarily returning a global optimum (Malinen & Fränti, 2014). Nevertheless, it returns a local optimum with respect to the SSE: the assignment step minimizes the SSE for a given set of centroids, while the update step minimizes the SSE for a given partition (Lin et al., 2019; Malinen & Fränti, 2014). The running time of one iteration is linear in the number of data points $n$ (Jain et al., 1999).

Among the advantages of this algorithm are its simplicity, time complexity and usability in a large area of subjects (Jain et al., 1999; Saini & Singh, 2015). However, despite its popularity, it also involves some drawbacks like the strong dependence on the initial choice of the cluster centroids (Fränti & Sieranoja, 2019; Saini & Singh, 2015) or its limitation to hyperspherical shaped clusters (Althoff, 2010). Another drawback, on which we focus in this paper, is its inability to control the number of objects contained in each cluster. Especially in high dimensional space and if many clusters are desired, often very small clusters seem to appear even if the data itself has a balanced distribution (Bradley et al., 2000).

In this paper, we refer to a *balanced clustering* as a clustering that distributes the data points evenly between the clusters. More formally, to incorporate situations in which the number of data points $n$ is not divisible by the number of clusters $k$, a balanced clustering requires that every cluster contains either $\left\lfloor \frac{n}{k} \right\rfloor$ or $\left\lceil \frac{n}{k} \right\rceil$ data points.

In general, balanced clustering is a *two-objective optimization problem* pursuing two goals that contradict each other: on the one hand, the SSE should be minimized, and on the other hand, the difference in cluster sizes should be minimized. If we were just interested in minimizing the SSE, we could apply an ordinary clustering algorithm like $k$-means, and if we were only interested in balancing, we could simply divide the data points randomly into clusters of the same size (Malinen & Fränti, 2014; Saini & Singh, 2015). Figure 1 demonstrates the case where minimizing the SSE with and without a balance constraint results in a different optimal clustering result.

To optimize both aims, there exist two different approaches, *hard-balanced*, also called *balance-constrained*, and *soft-balanced*, also called *balance-driven*, clustering. Both approaches differ in the way they assess the two objectives. Hard-balanced clustering strictly requires cluster size balance, whereas the minimization of the SSE serves as a secondary criterion. Soft-balanced clustering considers the balance of the cluster sizes as an aim but not as a mandatory requirement. It intends to find a compromise between the two goals, e.g.,

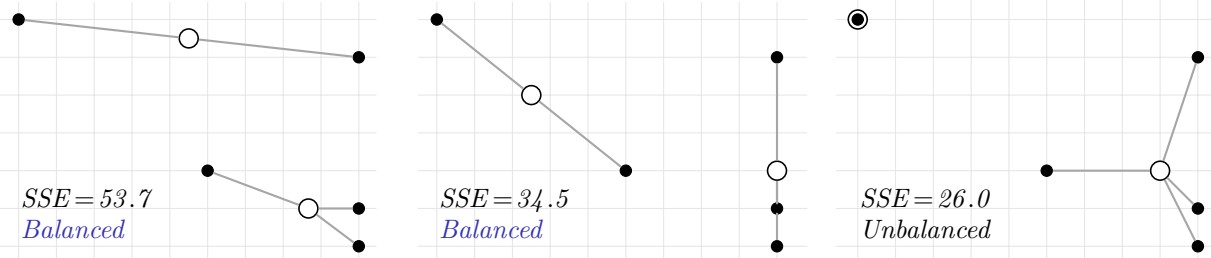

Figure 1: Balance constraints can lead to a different clustering result. There are two balanced clusterings with different SSE values (left and middle) and one unconstrained clustering optimized for SSE (right).

by weighting them or by using a heuristic which minimizes the SSE but indirectly creates more balanced clusters than the standard $k$-means algorithm. (Lin et al., 2019; Malinen & Fränti, 2014)

There exist a lot of applications for clustering that rely on a balanced distribution of the objects, i.e., a distribution in which every cluster contains exactly or approximately the same number of objects. Balanced clustering can be used in the division step of divide-and-conquer algorithms to provide equal sized partitions (Malinen & Fränti, 2014). In load balancing algorithms, balanced clustering can help to avoid unbalanced energy consumption in networking (Han et al., 2018; Liao et al., 2013; Lin et al., 2019) or to balance the workload of salesmen in the multiple travelling salesmen problem (Nallusamy et al., 2010). In the clustering of documents, articles or photos or in the creation of domain specific ontologies, balanced clustering can improve the resulting hierarchies by generating a more balanced view of the objects to facilitate the navigation and browsing (Banerjee & Ghosh, 2006). In retail chains, balanced clustering can be used to segment customers into equal sized groups to spend the same amount of marketing resources on each segment or to group similar products into categories of specified sizes to match units of shelf or floor space (Banerjee & Ghosh, 2006). Cost function leading to more balanced cluster sizes was used in Fränti et al. (2022) to allow manual investigation of the content of the diagnosis clusters.

In this paper, we propose a balanced clustering algorithm based on the k-means algorithm. Its main principle is an increasing penalty term, which is added to the assignment function of the $k$-means algorithm and favours objects to be assigned to smaller clusters. Because of the increasing penalty term, the resulting balance degree of a clustering is not determined by a rather non-intuitive parameter, but by the point of termination of the algorithm. In this way, the desired balance degree can be specified precisely and the algorithm can always be continued to ensure that it does not produce a better clustering with respect to the SSE satisfying the given balance requirement in future iterations.

The rest of the paper is organized as follows. We begin in Section 2 by summarizing the related work. In Section 3 we present the proposed algorithm. After demonstrating its main principle on a small exemplary data set, we focus on the magnitude of the penalty term and its computation. Further, we briefly consider its termination criterion and time complexity. Section 6 shows the results. We compare the proposed method to the regularized $k$-means algorithm by Lin et al. (2019) and the balanced $k$-means algorithm by Malinen & Fränti (2014) on several data sets.

## 2 Balanced Clustering

We next review the existing approaches for balanced clustering. Approaches for hard-balanced clustering are reviewed in Section 2.1 and for soft-balanced clustering in Section 2.2, respectively.

### 2.1 Hard-balanced Clustering

A popular approach to face this problem is to formulate the assignment step of the standard $k$-means algorithm as an optimization problem satisfying balance constraints and solve it by linear programming. The

*constrained k-means algorithm* proposed by Bradley et al. (2000) follows this approach and ensures clusters of given minimum sizes by solving a minimum cost flow problem. Depending on the chosen minimum cluster sizes, this algorithm can also be used for soft-balanced clustering. The time complexity of the assignment step is $\mathcal{O}(k^{3.5}n^{3.5})$, which makes the algorithm much slower than the standard $k$-means algorithm and reduces its scalability especially for large data sets (Malinen & Fränti, 2014).

The *balanced k-means algorithm* proposed by Malinen & Fränti (2014) solves the assignment step of $k$-means by the *Hungarian algorithm*. This reduces the time complexity of the assignment step compared to the previous method to $\mathcal{O}(n^3)$, but is still too slow for large data sets.

Another hard-balanced clustering algorithm formulating the assignment step of the standard $k$-means algorithm as a linear programming problem is the algorithm proposed by Tang et al. (2019). They claimed that the average time complexity of their algorithm is only $\mathcal{O}(mn^{1.65})$ to $\mathcal{O}(mn^{1.7})$ ($m$ denotes the number of iterations), which improves the running time remarkably compared to the above-mentioned algorithms.

An even faster algorithm was proposed by Zhu et al. (2010). This algorithm does not follow the iterative structure of the $k$-means algorithm but transforms the balanced clustering problem directly into a linear programming problem by using a heuristic function. However, this algorithm cannot keep up with the quality of the clustering achieved by the other algorithms (Tang et al., 2019).

A further method following the linear programming approach is *regularized k-means* by Lin et al. (2019). It extends the previous models by adding a balance regularization term to the objective function, that can be adapted according to the requirements. Thus, depending on the chosen regularization term, this algorithm can also be used for soft-balanced clustering.

Some further proposed algorithms for hard-balanced clustering follow different approaches. For example, the *neural gas clustering algorithm* is adapted by Luptáková et al. (2016) to handle given cluster sizes. *Conic optimization* is used by Rujeerapaiboon et al. (2019), which also allows to provide bounds on the suboptimality of the given solution. The *fuzzy c-means algorithm* is applied by Chakraborty & Das (2019) before using the resulting partial belongings and the given size constraints to finally assign the data points to the clusters. A *basic variable neighbourhood search heuristic* following the *less is more approach* was proposed by Costa et al. (2017). This heuristic performs a local descent method to explore neighbours, which are obtained by swapping points from different clusters in the current optimal solution. Recently Zhou et al. (2021) proposed a *memetic algorithm* combining a crossover operator to generate offspring and a responsive threshold search alternating between two different search procedures to optimize the solution locally. A *greedy randomized adaptive search procedure* combined with a strategic oscillation approach to alternate between feasible and infeasible solutions is used by Martín-Santamaría et al. (2022).

## 2.2 Soft-balanced Clustering

A popular approach for the soft-balanced clustering problem is the use of a multiplicative or additive bias in the assignment function of the standard $k$-means algorithm. First, Banerjee & Ghosh (2004) proposed to use the *frequency sensitive competitive learning* method. Competitive units, here clusters competing for data points, are penalized in proportion to the frequency of their winning, aiming at making all units participate. Banerjee & Ghosh (2004) applied this method by introducing a multiplicative bias term in the objective function of the standard $k$-means algorithm, which weights the distance between a data point and a centroid depending on the number of data points already assigned to the cluster. In this way, smaller clusters are favoured in the assignment step.

They also provided a theoretical background for their approach. The $k$-means algorithm implicitly assumes that the overall distribution of the data points can be decomposed into a mixture of isotropic Gaussians with uniform prior. Banerjee & Ghosh (2004) followed the idea to shrink the Gaussians in proportion to the number of data points that have been assigned to them by dividing the covariance matrix of each cluster by the number of data points assigned to it. Maximizing the log-likelihood of a data point with respect to this framework leads to the multiplicative bias (Althoff, 2010; Banerjee & Ghosh, 2004).

A similar approach was presented by Althoff et al. (2011). They also adapted the assumption that a data point is distributed according to a mixture of isotropic Gaussians with uniform prior. But instead of changing

the shape of the clusters by shrinking the Gaussians, they adjusted their prior probabilities such that they decrease exponentially in the number of data points assigned to them. Thus, the more data points a Gaussian contains, the lower its prior probability becomes. Maximizing the log-likelihood of a data point with respect to this framework results in an additive bias. Liu et al. (2018) complemented their work by providing the objective function and adding a theoretical analysis with respect to convergence and bounds in terms of bicriteria approximation.

Further algorithms use the *least square linear regression* method combined with a balance constraint that aims at minimizing the variance of the cluster sizes (Han et al., 2018; Liu et al., 2017). The least square regression error is minimized in each iteration such that the accuracy of the estimated hyperplanes, which partition the data into clusters, improves step by step.

Li et al. (2018) proposed an algorithm following the approach of the *exclusive lasso*. This method models a situation in which variables within the same group compete with each other (Zhou et al., 2010). They computed the exclusive lasso of the cluster indicator matrix, which equals the sum of the squared cluster sizes, and used it as a balance constraint by adding it as a bias to the objective function of the standard $k$-means algorithm.

A more generalized method, that can deal with different balance structures (cardinality, variance and density), was proposed by Gupta et al. (2018). In the assignment step of the standard $k$-means algorithm a multiplicative weight is added as a bias and additionally a balancing phase is introduced after each update step. In this phase, points are shifted from clusters with the largest weights to clusters with the lowest weights. The weights are updated after each iteration and their computation depends on the chosen balance structure.

A completely different approach, proposed by Banerjee & Ghosh (2006), introduced a stepwise working soft-balanced clustering algorithm, that provides, even though it is soft-balanced, some balance guarantees in form of minimum cluster sizes. First, a representative subset of the data points is sampled from the data, which then is clustered by an existing clustering algorithm. Since the amount of sampled points is small compared to the size of the data set, a slightly more complex algorithm can be chosen. Afterwards, the remaining points are distributed to the existing clusters respecting the balance constraints, and finally refinements are made to further minimize the objective function.

Lin et al. (2022) proposed an algorithm called *$\tau$-balanced clustering*. The variable $\tau$ denotes the maximal difference between the sizes of any two clusters and can be determined by the user. By setting this variable to one, a hard-balanced clustering algorithm is obtained. In the assignment step, a data point is assigned to the cluster whose centroid is closest if the size restrictions are not violated. Otherwise, it replaces a data point already assigned to the cluster if that point is farther from the centroid than the point that has to be assigned, else the cluster with the next nearest centroid is considered. After every data point is assigned to a cluster, the cluster centroids are updated according to the standard $k$-means algorithm. The algorithm terminates when the cluster centroids converge.

## 3 Proposed Method

We modify the standard $k$-means algorithm such that it does not only consider the squared distances to the cluster centroids, but also takes the cluster sizes into account. Our approach is to add a penalty term to the objective function of the standard $k$-means algorithm, which depends on the number of data points already assigned to a cluster. Thus, we want to create a new objective function in the shape of

$$\text{SSE}_{bal} = \text{SSE} + \sum_{j=1}^{k} penalty(j). \tag{2}$$

The penalty term of a cluster has to increase with the number of data points contained in that cluster. In this way, clusters containing few data points are favoured to get more points, and clusters containing many points are unfavoured. Thus, the clustering becomes more balanced. An easy example for a penalty term satisfying this property is the function that just returns the size of a cluster, $penalty(j) = n_j$, where $n_j$ denotes the number of data points already assigned to the $j^{th}$ cluster. Apparently, the use of this function as the penalty

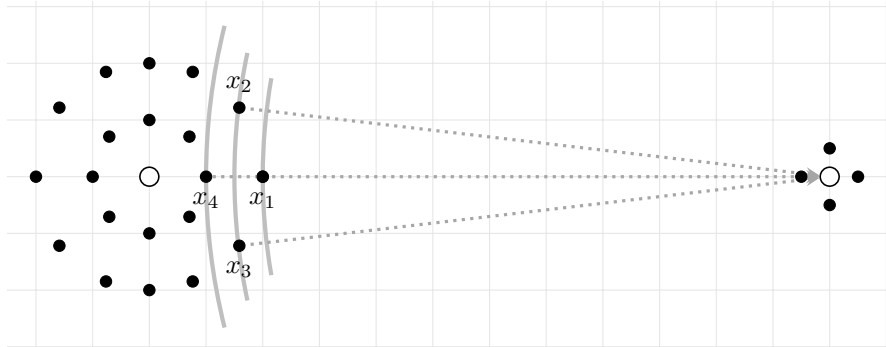

Figure 2: Intended behaviour of a gradually increasing penalty term

term is too general because it does not take the scale between the SSE and the number of data points into account.

If the penalty term returns constant zero, the optimization of 2 reduces to the minimization of the SSE, hence to the same problem that the standard $k$-means algorithm is facing.

The addition of a penalty term to the assignment function of the standard $k$-means algorithm follows the approach of (Liu et al., 2018) and (Althoff et al., 2011). The crucial point, in which we deviate from their method, is the determination of the penalty term. An appropriate value of the penalty term is essential for the algorithm in order to produce reasonable results, but it strongly depends on the given data set.

### 3.1 Increasing Penalty Term

We address this problem by introducing an increasing penalty term. The idea is to start off with a clustering produced by the standard $k$-means algorithm that yields a reasonably good clustering quality in minimizing the SSE and use a gradually increasing penalty term to shift the focus towards the balance. Figure 2 illustrates this intended behaviour: starting from the initial situation containing two clearly separated clusters, the penalty term increases such that first only point $x_1$ joins the cluster on the right, afterwards points $x_2$ and $x_3$, after that point $x_4$. In this way, the balance quality increases while the clustering quality decreases. The algorithm terminates as soon as the desired balance requirement is met.

Following this approach, we further avoid using a non-intuitive parameter that determines the trade-off between the clustering and the balance quality like many soft-balanced clustering algorithms do (Althoff et al., 2011; Li et al., 2018; Lin et al., 2019). Instead, the balance degree is determined by the time of termination, for which a criterion in form of a balance requirement can be given (see Section 5.2 for different ways to measure the balance quality). Thus, if, for example, a user wishes to reach a certain maximum difference between the smallest and the largest cluster, the algorithm will continue until such a clustering is reached. Therefore, this approach represents a very intuitive way for defining the trade-off between clustering and balance performance.

Formally, we are looking for a penalty term that not only depends on the size of a cluster, but also on the number of iterations. Therefore, we introduce the function *scale*, a strictly increasing function depending on the number of iterations, and define

$$penalty(j, iter) = scale(iter) \cdot n_j.$$

As the value of the function *scale* becomes steadily larger with the number of iterations, the influence of the cluster sizes increases relative to the error term SSE in each iteration. By introducing the dependency of the penalty term on the number of iterations, the objective function also becomes dependent on the number of iterations.

## 3.2 Scaling the Penalty Term

Probably one of the easiest ways to define the function *scale* consists in choosing it as a classical function like a logarithmic, linear, quadratic or exponential function. However, the problem with using one of these functions is the uniqueness of each data set. Using an increasing penalty term does not mean that we no longer have to be concerned about an appropriate magnitude of the penalty term.

If the penalty term is too small, it takes a very long time to reach a balanced clustering. On the other hand, if the term becomes too large too fast, overlapping clusters are produced. This can be seen as follows: in Figure 2, in the optimum case in one iteration the penalty term is large enough such that point $x_1$ changes to the smaller cluster on the right, but the term is not large enough to make the points $x_2$, $x_3$ and $x_4$ change to the cluster on the right. In the next iteration, the penalty term increases such that also the points $x_2$ and $x_3$ change to the cluster on the right, but still the penalty term is not large enough to make point $x_4$ change to the cluster on the right. In the next iteration, the penalty term further increases such that now also point $x_4$ changes to the cluster on the right. But if the penalty term increases too fast, for example, if points $x_1$ and $x_4$ can change to the cluster on the right in the same iteration for the first time, it can happen that $x_4$ changes to the cluster on the right, but $x_1$ does not. Then overlapping clusters have been formed.

Therefore, the size and growth of the penalty term are critical and due to the uniqueness of each data set, one data set may require a smaller penalty term, while other data sets need larger penalties and the required growth of the penalties can also differ.

Thus, we take a slightly more complicated approach than using a classical function, but also a more effective approach. The main principle is that progress in producing a more balanced clustering out of an unbalanced one happens only when a data point changes from a larger to a smaller cluster. Thus, the value of *scale* should be large enough to make a data point change from a larger to a smaller cluster. At the same time, it should not be chosen too large to preserve a good clustering quality, since a too fast increasing penalty term tends to lead to overlapping clusters.

This seems to be a high demand for the value of *scale*, but indeed, during one iteration of the algorithm we can compute the minimum value of the penalty term for the next iteration that is necessary to make at least one data point change to a smaller cluster.

Let us start simple: we assume that we are in iteration *iter* in the assignment phase of data point $x_i$ and its old cluster is denoted by $j_{old}$. Then, data point $x_i$ is able to change to cluster $j_{new}$ containing less data points, $n_{j_{new}} < n_{j_{old}}$, only if

$$
\begin{aligned}
& ||x_i - c_{j_{new}}||^2 + penalty(j_{new}, iter) < \quad ||x_i - c_{j_{old}}||^2 + penalty(j_{old}, iter) \\
\Leftrightarrow \quad & ||x_i - c_{j_{new}}||^2 - ||x_i - c_{j_{old}}||^2 \quad < \quad penalty(j_{old}, iter) - penalty(j_{new}, iter) \\
\Leftrightarrow \quad & ||x_i - c_{j_{new}}||^2 - ||x_i - c_{j_{old}}||^2 \quad < \quad scale(iter) \cdot n_{j_{old}} - scale(iter) \cdot n_{j_{new}} \\
\Leftrightarrow \quad & \frac{||x_i - c_{j_{new}}||^2 - ||x_i - c_{j_{old}}||^2}{n_{j_{old}} - n_{j_{new}}} \quad < \quad scale(iter).
\end{aligned}
\tag{3}
$$

Note that the last equivalence relies on our assumption

$$
n_{j_{new}} < n_{j_{old}}.
$$

First, if inequality 3 holds, data point $x_i$ can change to cluster $j_{new}$. This does not imply that $x_i$ will indeed change to cluster $j_{new}$, because there can be another cluster that leads to an even smaller cost term. However, point $x_i$ will change its cluster.

On the other hand, if inequality 3 does not hold, data point $x_i$ will not change to cluster $j_{new}$ during this iteration. Now the left-hand side of the inequality becomes interesting: it describes the value of *scale* which is at least necessary to enable the change of data point $x_i$ from cluster $j_{old}$ to cluster $j_{new}$. In other words, if we choose $scale(iter + 1)$ as this value, data point $x_i$ will be able to change to cluster $j_{new}$ during the next iteration (assuming that there are no changes in the assignments and locations of the clusters $j_{old}$ and $j_{new}$ until the assignment phase of $x_i$ in the next iteration). We denote this minimum value that is necessary to

enable the change of data point $x_i$ from its old cluster $j_{old}$ to cluster $j_{new}$, by $scale(iter, i, j_{new})$ and define it, based on inequality 3, as

$$scale(iter, i, j_{new}) = \begin{cases} \frac{||x_i - c_{j_{new}}||^2 - ||x_i - c_{j_{old}}||^2}{n_{j_{old}} - n_{j_{new}}} + \varepsilon & \text{if } n_{j_{old}} > n_{j_{new}} \\ \infty & \text{otherwise,} \end{cases} \tag{4}$$

where $\varepsilon$ is a very small number $> 0$ (to account for the inequality in 3), $j_{old}$ denotes the old cluster of $x_i$, and the cluster assignments and centroid locations at the time of the assignment phase of data point $x_i$ during iteration *iter* are used.

During the iteration over all data points and clusters, we save the minimum of all these values which are larger than the current value $scale(iter)$ and denote it by $scale_{min}(iter)$, i.e.,

$$scale_{min}(iter) = \min_{\substack{i \in \{1, \dots, n\}, \\ j_{new} \in \{1, \dots, k\}}} \big\{ scale(iter, i, j_{new}) \mid scale(iter, i, j_{new}) > scale(iter) \big\}. \tag{5}$$

We only consider values of $scale(iter, i, j_{new})$ which are larger than the current value $scale(iter)$, used in the penalty term in this iteration, because if $scale(iter + 1)$, the value that will be used in the next iteration, is smaller than the current one, it enables data points which already changed from a larger to a smaller cluster to change back to the larger cluster again. In this case, we take a step backwards concerning our aim to balance the cluster sizes.

The value $scale_{min}(iter)$ is the minimum value that is needed for *scale* in the next iteration to make a data point change from a larger to a smaller cluster, hence, it is the value we are looking for. Therefore, we require

$$scale(iter + 1) \geq scale_{min}(iter). \tag{6}$$

### 3.3 Increasing Penalty Factor f

Perhaps one expected an equality instead of an inequality in 6. The problem of using the equality is that the closer we choose the value of $scale(iter + 1)$ to the value of $scale_{min}(iter)$, the less data points will change their clusters during iteration $iter + 1$. In other words, if we choose $scale(iter + 1)$ exactly as $scale_{min}(iter)$, in many iterations only one data point will change from a larger to a smaller cluster. This is not that bad if the cluster sizes are almost balanced and there are only few data points left that have to change the clusters in order to obtain a balanced clustering, but if the cluster sizes are far from being balanced, then the algorithm takes a lot of iterations until a balanced clustering is reached.

On the other hand, if $scale(iter + 1)$ is chosen much larger than $scale_{min}(iter)$, a lot of data points are able to change their clusters during the iteration $iter + 1$, and the probability that *wrong* data points are assigned to the smaller clusters, such that overlapping clusters are produced, increases. Thus, the choice of the relation between $scale(iter + 1)$ and $scale_{min}(iter)$ seems to be critical, and this decision indeed results in a trade-off between the clustering quality and the running time. For now, we deal with this problem by introducing the increasing penalty factor $f$ and define

$$scale(iter + 1) = f \cdot scale_{min}(iter) \text{ where } f \geq 1. \tag{7}$$

### 3.4 Partly Remaining Fraction c

In the standard $k$-means algorithm, the cluster sizes do not need to be known during the assignment step of a data point because they are not necessary for the computation of the cost of assigning a data point to a cluster. However, if we include a penalty term as indicated in Equation 2, we have to know the cluster sizes during the assignment of a data point.

The calculation of the cluster sizes itself is quite simple, but the crucial question is when to update the cluster sizes. This could take place after every complete iteration of the algorithm, after the assignment phase of every data point or before and after the assignment phase of every data point (in the sense of removing a data point from its cluster before its assignment starts). Indeed, none of these approaches is really good, each

Table 1: Recommended ranges and default values for the parameters used in BKM+

| Parameter | Recommended range | Default value |
|:---:|:---:|:---:|
| $c$ | 0.15 - 0.20 | 0.15 |
| $f$ | 1.01 - 1.10 | Function $f_{iter}$ depending on the number of iterations with $f_{iter}(1) = 1.10$ and $f_{iter}(i) = 1.01$ for $i > 100$, linear interpolation for $1 < i \leq 100$ |

has its shortcomings, like producing oscillating or overlapping clusters. The most promising approach, which we choose, is the following. A data point is removed partially from its old cluster before its assignment and only after its new assignment it is removed completely from its old cluster and is added to its new cluster. In this way, during the assignment of a data point, the point only partially belongs to its old cluster. To define *partially*, we introduce the constant $c$ with $0 < c < 1$, which we refer to as the partly remaining fraction of a data point.

## 4    Algorithm BKM+

A first version of the proposed algorithm is presented in Algorithm 1. We call it *balanced k-means revisited* (BKM+).

In each iteration of the algorithm, each data point is assigned to the cluster $j$ which minimizes the cost term $||x_i - c_j||^2 + p_{now} \cdot n_j$. In this process, the data point is first partly removed from its old cluster, see function REMOVEPOINT, and afterwards added to its new cluster and completely removed from its old cluster, see function ADDPOINT. Directly after the computation of the cost term, the variable $p_{next,i}$, corresponding to $\min_{j_{new} \in \{1,...,k\}} \{scale(iter, i, j_{new})\}$, is computed using the function PENALTYNEXT, which implements Equation 4. If this number is a candidate for $p_{next}$, its value is taken. After all data points are assigned, the penalty term $p_{now}$ is set for the following iteration by multiplying $p_{next}$ by the increasing penalty factor $f$. Afterwards, the assignments of the data points start again. The algorithm terminates as soon as the largest cluster is at most $\Delta n_{max}$ data points larger than the smallest cluster.

### 4.1   Optimization of Parameters c and f

The algorithm still contains two parameters which have to be determined: the partly remaining fraction $c$ for defining the fraction of a data point that belongs to its old cluster during its assignment, and the increasing penalty factor $f$, which determines the factor by which the minimum penalty that is necessary in the next iteration in order to make a data point change to a smaller cluster is multiplied.

For $c$, a value between 0.15 and 0.20 is reasonable in every situation. The choice of $f$ is a trade-off between the clustering quality and the running time: if the focus is on the clustering quality, $f$ should be chosen as 1.01 or smaller, whereas if the running time is critical, setting $f$ to 1.05 or 1.10 is the better choice. A compromise is to choose $f$ as a function depending on the number of iterations. In the beginning $f$ can be chosen larger to ensure a fast progress of the penalty term, while in the end $f$ should be chosen smaller to avoid a negative influence on the clustering quality. For example, we can define a function $f_{iter}$, whose value is 1.10 in the first iteration and 1.01 starting from the 101[st] iteration, and between these iterations we linearly interpolate the values. Table 1 summarizes the recommended ranges and the default values for $c$ and $f$.

### 4.2   Application of the Standard $k$-means Algorithm

Until now, we applied the standard $k$-means algorithm in the beginning of the algorithm to obtain a clustering as a starting point since the computation of the penalty term relies on the cluster sizes. However, this raises the question whether it is necessary to apply the standard $k$-means algorithm until its termination.

---

**Algorithm 1** Balanced $k$-means revisited (preliminary)

---

INPUT: Data set containing $n$ data points $\{x_1, \ldots, x_n\}$, number of clusters $k$, partly remaining fraction $c$, increasing penalty factor $f$, maximum difference in cluster sizes $\Delta n_{max}$

OUTPUT: Balanced partition $\{p_1, \ldots, p_k\}$ of the data points

1: Apply the standard $k$-means algorithm to get an initial partition $\{p_1, \ldots, p_k\}$ and initial centroid locations $c_1, \ldots, c_k$
2: Set initial cluster sizes $n_1 \leftarrow |p_1|, \ldots, n_k \leftarrow |p_k|$
3: Set initial values of *scale* to $p_{now} \leftarrow 0$ and $p_{next} \leftarrow \infty$
4: Set initial values of the min and max cluster sizes to $n_{min} \leftarrow 0$ and $n_{max} \leftarrow n$
5: $iter \leftarrow 0$
6: **while** $n_{max} - n_{min} \leq \Delta n_{max}$ **do**
7:     **for each** $x_i$ **do**
8:         $j^- \leftarrow$ old cluster of $x_i$
9:         REMOVEPOINT$(i, j^-)$
10:         $j^+ \leftarrow \arg\min_{j \in \{1,\ldots,k\}} (||x_i - c_j||^2 + p_{now} \cdot n_j)$
11:         $p_{next,i} \leftarrow \min_{j \in \{1,\ldots,k\}}$ PENALTYNEXT$(i, j^-, j)$
12:         **if** $p_{now} < p_{next,i} < p_{next}$ **then**
13:             $p_{next} \leftarrow p_{next,i}$
14:         ADDPOINT$(i, j^+, j^-)$
15:     $n_{min} \leftarrow \min_{j \in \{1,\ldots,k\}}(n_j)$, $n_{max} \leftarrow \max_{j \in \{1,\ldots,k\}}(n_j)$
16:     $p_{now} \leftarrow f \cdot p_{next}$, $p_{next} \leftarrow \infty$
17:     $iter \leftarrow iter + 1$
18: **return** $\{p_1, \ldots, p_k\}$

19: **function** REMOVEPOINT$(i, j^-)$
20:     $p_{j^-} \leftarrow p_{j^-} \setminus \{x_i\}$
21:     $c_{j^-} \leftarrow \frac{1}{|p_{j^-}|} \sum_{x_i \in p_{j^-}} x_i$
22:     $n_{j^-} \leftarrow n_{j^-} - 1 + c$

23: **function** ADDPOINT$(i, j^+, j^-)$
24:     $p_{j^+} \leftarrow p_{j^+} \cup \{x_i\}$
25:     $c_{j^+} \leftarrow \frac{1}{|p_{j^+}|} \sum_{x_i \in p_{j^+}} x_i$
26:     $n_{j^+} \leftarrow n_{j^+} + 1$
27:     $n_{j^-} \leftarrow n_{j^-} - c$

28: **function** PENALTYNEXT$(i, j^-, j)$
29:     **if** $n_{j^-} > n_j$ **then**
30:         **return** $\frac{||x_i - c_j||^2 - ||x_i - c_{j^-}||^2}{n_{j^-} - n_j}$
31:     **else**
32:         **return** $\infty$

---

After the initialization of the cluster centroids, one iteration of the standard $k$-means algorithm is necessary to assign the data points to the clusters and another iteration is necessary to compute the starting value of $scale$. Since an iteration of the standard $k$-means algorithm corresponds to an iteration of our algorithm setting $scale = 0$, in the beginning we can perform two iterations of the standard $k$-means algorithm by defining $scale(0) = scale(1) = 0$. Afterwards we can compute the values of $scale(iter)$ based on the values of $scale_{min}(iter - 1)$ like we defined in Equations 4, 5 and 7. There are no meaningful differences in the clustering quality if the standard $k$-means algorithm is applied for only two iterations in the beginning, but the running time is significantly faster.

### 4.3 Computation of the Function scale

We introduced the function $scale$, which returns the minimum value of the penalty term that is necessary to make at least one data point change its cluster multiplied by the increasing penalty factor $f$. To compute its value for the iteration $iter + 1$, we have to compute the values $scale(iter, i, j)$ for all $n \cdot k$ combinations of data points and clusters.

However, the use of the function $scale$ does not require computing the cost of the assignment and the value of $scale(iter, i, j)$ for every of the $n \cdot k$ combinations of data points and clusters as it may look in Algorithm 1. The information contained in these values is different, but it is not independent. One possibility to reduce the number of calculations is to always compute $scale(iter, i, j)$ and compare it to the value of $scale(iter)$. By the result of this comparison we know whether data point $x_i$ could move to cluster $j$ and only in this case we have to compute the cost of assigning $x_i$ to $j$.

### 4.4 Termination Criterion

In Algorithm 1 we used $\Delta n_{max}$, the maximum difference in cluster sizes, as the termination criterion. Since the desired balance degree is application dependent, we keep the termination criterion as a user input, but in a generalized way: instead of the maximum difference in cluster sizes, an arbitrary balance criterion can be selected, e.g., the standard deviation in cluster sizes (SDCS) or the normalized entropy ($N_{entro}$) (for an explanation of these measures see Section 5.2).

In some situations it can be advantageous to continue running the algorithm even if the balance criterion is already met and to return the clustering with the best clustering quality satisfying this criterion. This approach requires saving the best clustering reached so far, but it can help to improve the clustering quality. Especially if the data set is not well known (in the sense that the balance of the clustering producing the minimum SSE on the data set is not approximately known), this approach prevents a resulting clustering from being optimizable in both the clustering and the balance quality. Additionally, the current penalty term can be kept for the next iteration if the current clustering already meets the balance criterion and its SSE is the lowest found so far satisfying this criterion.

### 4.5 Time Complexity

The running time of the algorithm depends on the number of iterations and the running time per iteration.

In the first part of each iteration, the iteration over all $n$ data points and $k$ clusters dominates. Its complexity is $\mathcal{O}(nk)$. In the second part of each iteration, the dominating operation is the iteration over the clusters to compute the balance measure which takes $\mathcal{O}(k)$. If the SSE is computed or the current assignments are saved, the time complexity for the second part increases to $\mathcal{O}(n)$. However, independent of computing the SSE or saving the assignments in the second part of each iteration, the complexity of the first part dominates. Therefore, the complexity of one iteration of the algorithm is $\mathcal{O}(nk)$.

The number of iterations is difficult to predict. It primarily depends on the chosen data set and termination criterion, but also on the increasing penalty factor $f$. For example, to reach a balance clustering, on the data set wine (for properties and references of the mentioned data sets see Section 5) the algorithm takes about 15 iterations setting $c = 0.15$ and $f = 1.01$, on the data set S3 it takes about 90 iterations setting $c = 0.1$ and $f$ to the function $f_{iter}$, on the data set A3 it takes about 180 iterations setting $c = 0.2$ and $f$ to the function

$f_{iter}$, on the data set unbalance it takes about 400 iterations setting $c = 0.01$ and $f = 1.01$, and on the data set birch1 it takes about 600 iterations setting $c = 0.15$ and $f$ to the function $f_{iter}$.

## 5 Experimental Setup

The balanced clustering algorithms we chose for comparison are regularized $k$-means (RKM) proposed by Lin et al. (2019) and balanced $k$-means (BKM) by Malinen & Fränti (2014). RKM is an algorithm that can be used for both soft- and hard-balanced clustering depending on the selected balance regularization term, while BKM is solely a hard-balanced clustering algorithm. Another method considered for comparison was $\tau$-balanced clustering by Lin et al. (2022). Unfortunately, the publicly available implementation was incomplete.

For our method we use the proposed algorithm with all the mentioned optimizations. The partly remaining fraction $c$ is set to 0.15 and for the increasing penalty factor $f$ the function $f_{iter}$ is used. The algorithm is implemented in C++[1].

The source codes of BKM[2] and RKM[3] are publicly available. BKM is implemented in MATLAB, while RKM is implemented in C++. For both algorithms we used the default settings, unless specified otherwise. As a platform, Intel Core i5-7300U 2.60GHz processor with 8GB memory was used.

In the experiments we considered the artificial data sets S1-S4, A1-A3, birch1, birch2, unbalance and dim32 (Fränti & Sieranoja, 2018; Rezaei & Fränti, 2016; Zhang et al., 1997) and the real-world data sets vowel recognition, iris, user knowledge, wine, ionosphere, libra and multiple features from the UCI machine learning repository (Dua & Graff, 2017; Kahraman et al., 2013). Detailed information of the data sets is shown in the first column of Table 2. The first number under the name of a data set denotes its size, the second number its dimension and the third number refers to the number of clusters sought in the data set.

### 5.1 Clustering Quality

There are many ways to assess the quality of a clustering (Rezaei & Fränti, 2016). We use the sum of squared error function (SSE), defined in Equation 1, as an internal evaluation criterion, and the normalized mutual information (NMI) as an external criterion (Liu et al., 2017; Strehl & Ghosh, 2002; Strehl et al., 2000). The NMI of a labelling with $k$ different labels and a clustering with $k$ clusters is defined as

$$\text{NMI} = \frac{\sum_{h=1}^{k} \sum_{l=1}^{k} n_{h,l} \cdot \log\left(\frac{n \cdot n_{h,l}}{n_h \cdot n_l}\right)}{\sqrt{\left(\sum_{h=1}^{k} n_h \cdot \log\left(\frac{n_h}{n}\right)\right)\left(\sum_{l=1}^{k} n_l \cdot \log\left(\frac{n_l}{n}\right)\right)}},$$

where $n_h$ denotes the number of objects in class $h$, $n_l$ refers to the number of objects assigned to cluster $l$, $n_{h,l}$ defines the number of objects which are in class $h$ as well as assigned to cluster $l$, and $n$ denotes the total number of objects. A value of one means a perfect match between the labelling and the clustering, whereas a value close to zero indicates a random partitioning (Liu et al., 2017).

### 5.2 Balance

A popular and intuitive approach to measure the balancing behaviour of a clustering algorithm is the standard deviation in cluster sizes (SDCS). Let $n_j$ denote the size of the $j^{\text{th}}$ cluster and $k$ the number of clusters, then the SDCS is defined as

$$\text{SDCS} = \sqrt{\frac{1}{k-1} \sum_{j=1}^{k} \left(n_j - \frac{n}{k}\right)^2}. \tag{8}$$

---

[1]The source code of BKM+ can be found at `https://cs.uef.fi/ml/software`.
[2]The source code of BKM can be found at `https://cs.uef.fi/ml/software`.
[3]The source code of RKM can be found at `https://github.com/zhu-he/regularized-k-means`.

The smaller the standard deviation becomes the better the balance. A value of zero implies a perfectly balanced clustering (note that if $n$ is not divisible by $k$, a value of zero will never be reached). (Althoff, 2010; Banerjee & Ghosh, 2004; Banerjee & Ghosh, 2006; Chakraborty & Das, 2019; Gupta et al., 2018)

Another way to measure the balance of a clustering is to consider the entropy of the distribution of the cluster sizes. To simplify comparisons, we again consider a normalized version of the entropy, which is formally defined as

$$\mathrm{N_{entro}} = -\frac{1}{\log(k)} \sum_{j=1}^{k} \frac{n_j}{n} \log\left(\frac{n_j}{n}\right). \tag{9}$$

A normalized entropy of one implies a perfectly balanced clustering, while a value of zero represents an extremely unbalanced clustering (again, the divisibility of $n$ by $k$ must be taken into account). (Han et al., 2018; Liu et al., 2017; Liu et al., 2018)

In addition to these measures that give a review of the overall distribution of the cluster sizes, it can be useful to know whether a clustering contains extremely small clusters. To check this undesirable behaviour, the minimum cluster size can be considered. (Althoff, 2010; Banerjee & Ghosh, 2004; Banerjee & Ghosh, 2006)

## 6 Results

We next present the experimental results. We consider the hard-balanced version of the proposed algorithm in Section 6.1 and the soft-balanced version in Section 6.2. The results are discussed in Section 6.3.

### 6.1 Hard-balanced Clustering

We start by comparing the hard-balanced version of the proposed algorithm to BKM and the hard-balanced version of RKM. For this purpose, we set the termination criterion of the proposed method to a maximum difference of one in the cluster sizes. BKM is a hard-balanced clustering algorithm by definition. To get a hard-balanced version of RKM we choose the regularization term for hard-balanced clustering proposed by the authors (Lin et al., 2019).

We run each algorithm 100 times on each data set and report the best and the mean sum squared error (SSE) and normalized mutual information (NMI) as well as the average running time. Since the resulting clustering depends on the initialization of the clusters and, in case of the proposed method, also on the order of the data points in the data set, every algorithm starts with the same initial assignments and orders of the data set. The results are shown in Table 2.

The results of all three methods are quite similar both regarding to SSE and NMI. Especially in case of the artificial data sets, the results are almost identical, both in mean and best run. Overall, RKM leads to slightly lower SSE values, but the difference is not significant. The main difference in performance is in the running time, in which both RKM and BKM+ are an order of magnitude faster than BKM. For example, the data sets birch1 and birch2 require > 12 hours by BKM whereas RKM and BKM+ requires only 1-2 minutes. The proposed algorithm BKM+ appears to be slightly faster than RKM if clusters are more overlapping, see data sets S1-S4.

In summary, regarding the clustering quality, on most of the data sets both RKM and BKM are superior to BKM+. However, if the application is time-critical, the proposed algorithm provides an alternative to at least BKM. For low dimensional data sets the proposed algorithm mostly also takes less time than RKM.

Table 2: Results for the SSE, NMI and running time if a balanced clustering is required, $n$ denotes the size of the data set, $d$ its dimension and $k$ refers to the number of clusters sought in the data set

| Data set $(n, d, k)$ | Algorithm | SSE Best | SSE Mean | NMI Best | NMI Mean | Time (sec) |
|---|---|---|---|---|---|---|
| S1 (5000, 2, 15) | BKM | 1.093 e+13 | one run | 0.947 | one run | 5346.56 |
| | RKM | 1.089 e+13 | 1.089 e+13 | 0.948 | 0.948 | 0.18 |
| | BKM+ | 1.095 e+13 | 1.100 e+13 | 0.948 | 0.946 | 0.19 |
| S2 (5000, 2, 15) | BKM | 1.433 e+13 | one run | 0.919 | one run | 4486.51 |
| | RKM | 1.428 e+13 | 1.428 e+13 | 0.921 | 0.921 | 0.20 |
| | BKM+ | 1.435 e+13 | 1.449 e+13 | 0.922 | 0.918 | 0.17 |
| S3 (5000, 2, 15) | BKM | 1.736 e+13 | one run | 0.795 | one run | 6890.88 |
| | RKM | 1.734 e+13 | 1.734 e+13 | 0.797 | 0.796 | 0.23 |
| | BKM+ | 1.736 e+13 | 1.737 e+13 | 0.799 | 0.796 | 0.13 |
| S4 (5000, 2, 15) | BKM | 1.652 e+13 | one run | 0.729 | one run | 6937.59 |
| | RKM | 1.651 e+13 | 1.651 e+13 | 0.729 | 0.729 | 0.27 |
| | BKM+ | 1.651 e+13 | 1.652 e+13 | 0.732 | 0.730 | 0.14 |
| A1 (3000, 2, 20) | BKM | 1.221 e+10 | one run | 0.984 | one run | 620.92 |
| | RKM | 1.221 e+10 | 1.221 e+10 | 0.984 | 0.984 | 0.14 |
| | BKM+ | 1.222 e+10 | 1.224 e+10 | 0.985 | 0.983 | 0.09 |
| A2 (5250, 2, 35) | BKM | 2.037 e+10 | one run | 0.989 | one run | 2191.33 |
| | RKM | 2.037 e+10 | 2.037 e+10 | 0.989 | 0.989 | 0.48 |
| | BKM+ | 2.038 e+10 | 2.048 e+10 | 0.989 | 0.986 | 0.31 |
| A3 (7500, 2, 50) | BKM | 2.905 e+10 | one run | 0.991 | one run | 14037.61 |
| | RKM | 2.905 e+10 | 2.905 e+10 | 0.991 | 0.991 | 1.08 |
| | BKM+ | 2.908 e+10 | 2.941 e+10 | 0.991 | 0.984 | 0.97 |
| birch1 (100000, 2, 100) | BKM | - | - | - | - | > 12h |
| | RKM | 9.288 e+13 | 9.288 e+13 | 0.989 | 0.989 | 106.20 |
| | BKM+ | 9.293 e+13 | 9.307 e+13 | 0.986 | 0.983 | 61.33 |
| birch2 (100000, 2, 100) | BKM | - | - | - | - | > 12h |
| | RKM | 4.569 e+11 | 4.569 e+11 | 0.999 | 0.999 | 80.61 |
| | BKM+ | 4.578 e+11 | 11.477 e+11 | 0.999 | 0.933 | 94.04 |
| unbalance (6500, 2, 8) | BKM | - | - | - | - | > 12h |
| | RKM | 1.700 e+13 | 1.700 e+13 | 0.632 | 0.632 | 0.25 |
| | BKM+ | 1.868 e+13 | 1.932 e+13 | 0.438 | 0.354 | 0.60 |
| vowel recognition[4] (871, 3, 6) | BKM | 3.314 e+7 | 3.324 e+7 | - | - | 64879.15 e−3 |
| | RKM | 3.314 e+7 | 3.314 e+7 | - | - | 13.19 e−3 |
| | BKM+ | 3.315 e+7 | 3.326 e+7 | - | - | 6.76 e−3 |
| iris (150, 4, 3) | BKM | 8.137 e+1 | 8.137 e+1 | 0.777 | 0.777 | 352.85 e−3 |
| | RKM | 8.137 e+1 | 8.137 e+1 | 0.777 | 0.777 | 0.44 e−3 |
| | BKM+ | 8.137 e+1 | 8.139 e+1 | 0.803 | 0.777 | 0.18 e−3 |
| user knowledge (403, 5, 4) | BKM | 7.023 e+1 | 7.114 e+1 | 0.441 | 0.298 | 10301.52 e−3 |
| | RKM | 7.022 e+1 | 7.088 e+1 | 0.396 | 0.296 | 3.52 e−3 |
| | BKM+ | 7.021 e+1 | 7.083 e+1 | 0.413 | 0.320 | 1.78 e−3 |

---

[4]There were no class labels available for the data set vowel recognition.

Table 2: Results for the SSE, NMI and running time if a balanced clustering is required, $n$ denotes the size of the data set, $d$ its dimension and $k$ refers to the number of clusters sought in the data set

| Data set $(n, d, k)$ | Algorithm | SSE | | NMI | | Time (sec) |
|---|---|---|---|---|---|---|
| | | Best | Mean | Best | Mean | |
| wine | BKM | $2.962\,\mathrm{e}{+}6$ | $2.979\,\mathrm{e}{+}6$ | 0.397 | 0.392 | $373.95\,\mathrm{e}{-}3$ |
| (178, 13, 3) | RKM | $2.962\,\mathrm{e}{+}6$ | $2.962\,\mathrm{e}{+}6$ | 0.397 | 0.397 | $0.54\,\mathrm{e}{-}3$ |
| | BKM+ | $2.962\,\mathrm{e}{+}6$ | $2.980\,\mathrm{e}{+}6$ | 0.429 | 0.399 | $0.92\,\mathrm{e}{-}3$ |
| dim32 | BKM | $2.325\,\mathrm{e}{+}5$ | $2.325\,\mathrm{e}{+}5$ | 1.000 | 1.000 | 18.46 |
| (1024, 32, 16) | RKM | $2.325\,\mathrm{e}{+}5$ | $2.325\,\mathrm{e}{+}5$ | 1.000 | 1.000 | 0.03 |
| | BKM+ | $2.325\,\mathrm{e}{+}5$ | $2.325\,\mathrm{e}{+}5$ | 1.000 | 1.000 | 0.10 |
| ionosphere | BKM | $2.434\,\mathrm{e}{+}3$ | $2.435\,\mathrm{e}{+}3$ | 0.105 | 0.104 | $4428.93\,\mathrm{e}{-}3$ |
| (351, 34, 2) | RKM | $2.434\,\mathrm{e}{+}3$ | $2.434\,\mathrm{e}{+}3$ | 0.105 | 0.105 | $0.86\,\mathrm{e}{-}3$ |
| | BKM+ | $2.434\,\mathrm{e}{+}3$ | $2.442\,\mathrm{e}{+}3$ | 0.120 | 0.104 | $3.83\,\mathrm{e}{-}3$ |
| libra | BKM | $6.459\,\mathrm{e}{+}7$ | $6.519\,\mathrm{e}{+}7$ | 0.314 | 0.247 | $11189.76\,\mathrm{e}{-}3$ |
| (360, 90, 15) | RKM | $6.491\,\mathrm{e}{+}7$ | $6.554\,\mathrm{e}{+}7$ | 0.278 | 0.240 | $14.24\,\mathrm{e}{-}3$ |
| | BKM+ | $6.395\,\mathrm{e}{+}7$ | $6.475\,\mathrm{e}{+}7$ | 0.156 | 0.125 | $69.90\,\mathrm{e}{-}3$ |
| multiple features | BKM | $1.768\,\mathrm{e}{+}6$ | one run | 0.750 | one run | 669.26 |
| (2000, 240, 10) | RKM | $1.750\,\mathrm{e}{+}6$ | $1.758\,\mathrm{e}{+}6$ | 0.796 | 0.735 | 0.22 |
| | BKM+ | $1.750\,\mathrm{e}{+}6$ | $1.758\,\mathrm{e}{+}6$ | 0.786 | 0.715 | 1.08 |

## 6.2 Soft-balanced Clustering

Next, we compare the soft-balanced version of the proposed algorithm to the soft-balanced version of RKM. For the comparison of the balance we use three different balance measures, the standard deviation in cluster sizes (SDCS), the normalized entropy ($N_{entro}$) and the minimum cluser size (for a definition of these measures see Section 5.2).

According to the balance measure, we set the termination criterion of the proposed method as described in Section 4.4 to a maximum value of SDCS, a minimum value of $N_{entro}$ or a minimum value of the minimum cluster size. To obtain clusterings of different balance degrees we vary these values. RKM optimizes the balance using a regularization term including a balance parameter $\lambda$. Depending on the balance measure that has to be optimized, the authors propose different regularization terms (Lin et al., 2019). We use the regularization term corresponding to the chosen balance measure and vary the parameter $\lambda$. As in the hard-balanced case, we run each algorithm 100 times with all balance settings on each data set using the same initializations. The mean of the SSE, the balance measure and the running time were reported.

Figures 3 and 4 show the results for the data sets S2 and S4. The plots on the left side show the results as a function of the balance measure (x-axis) and the SSE (y-axis), whereas the plots on the right side show the results as a function of the balance measure (x-axis) and the running time (y-axis). The SSE-results of the two methods are almost identical between RKM and BKM+ when the balance is given high emphasis and is measured by SDCS or $N_{entro}$. RKM provides slightly better results if the balance is measured by the minimum cluster size.

When the focus is more on the clustering quality instead of the balance quality, BKM+ is more stable and provides better SSE-results while the results of RKM start to deteriorate. The difference in running time becomes also significant and favours BKM+. Its running time is almost constant regardless of whether the focus is on high clustering quality or balance quality. The processing time of RKM becomes significantly higher when the focus is shifted towards the clustering quality.

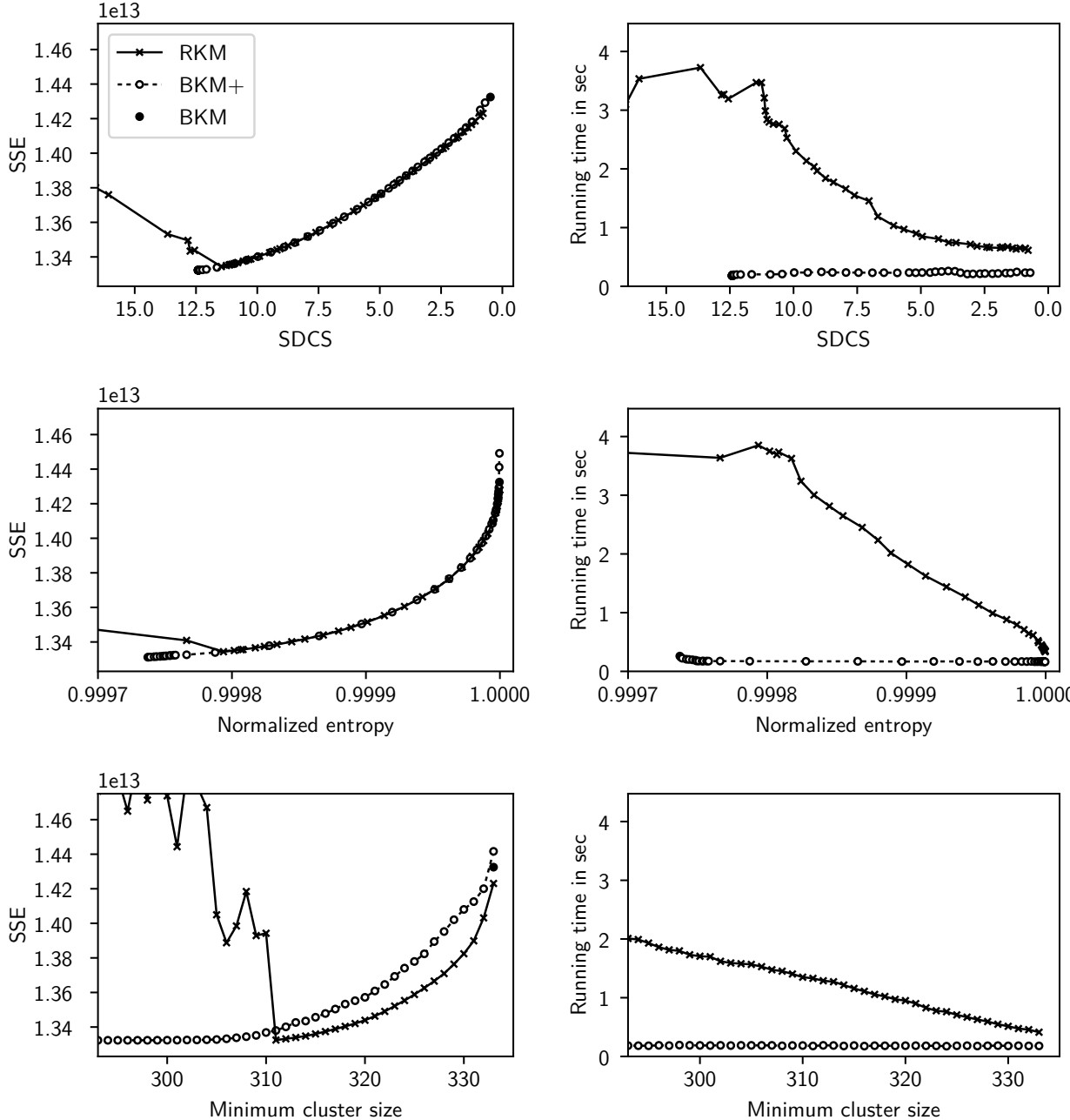

Figure 3: Results soft-balanced clustering for the data set S2

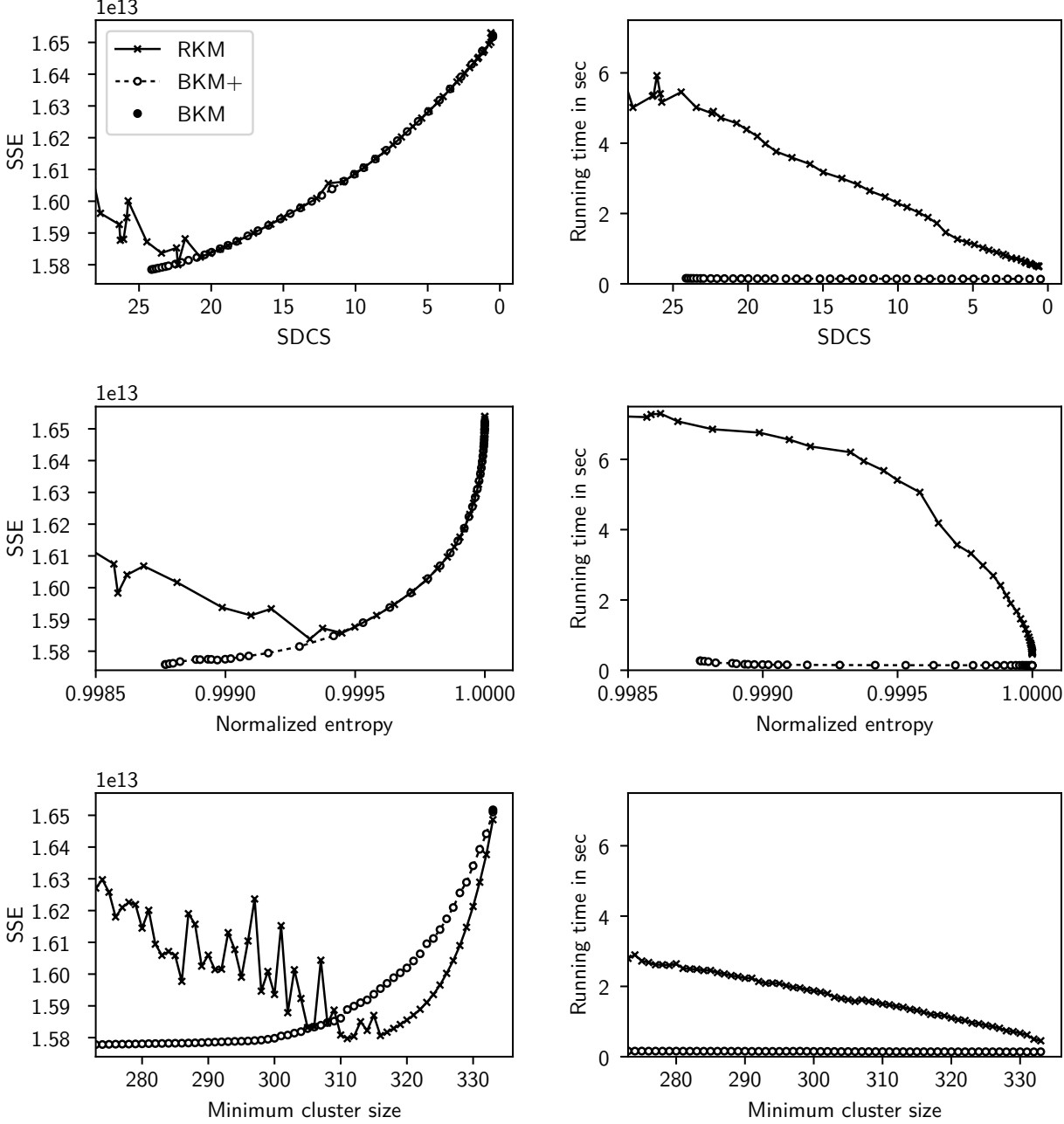

Figure 4: Results soft-balanced clustering for the data set S4

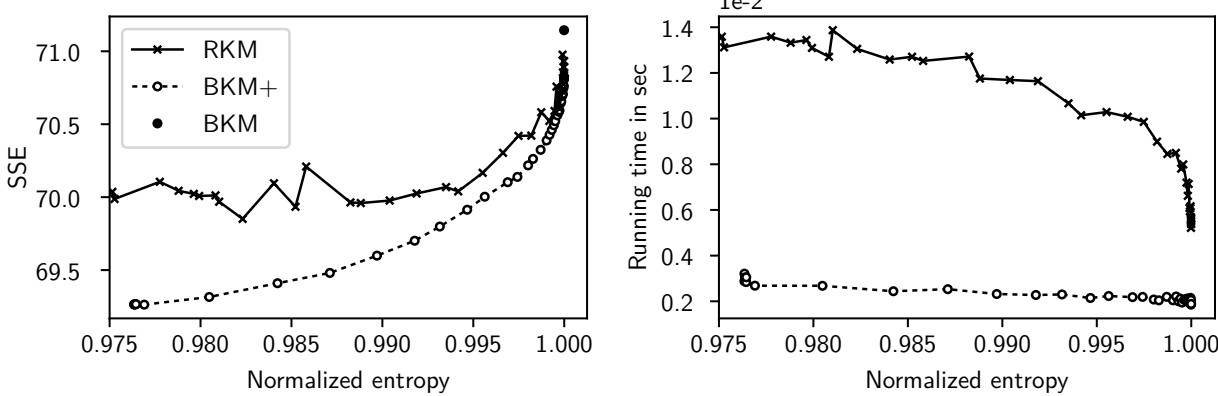

Figure 5: Results soft-balanced clustering for the data set user knowledge

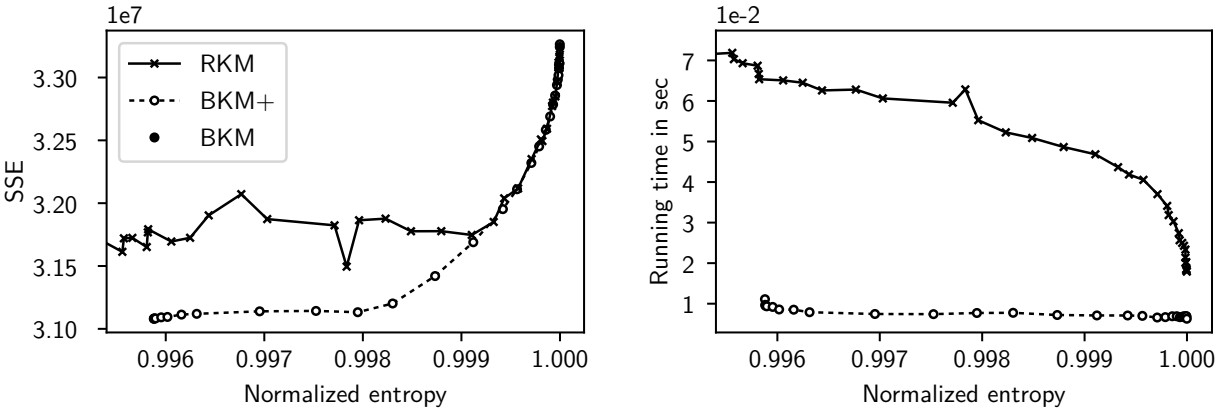

Figure 6: Results soft-balanced clustering for the data set vowel recognition

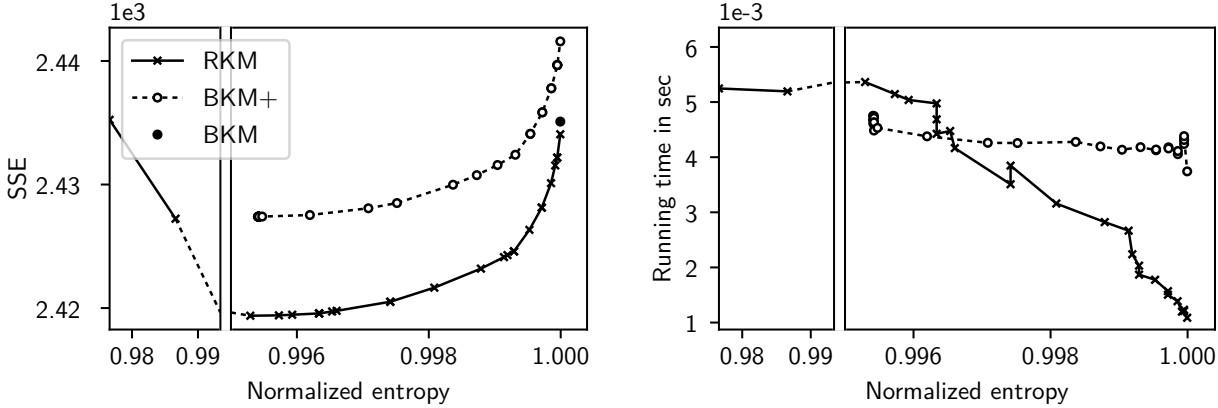

Figure 7: Results soft-balanced clustering for the data set ionosphere

Table 3: Results for SSE and running time for a clustering with $N_{entro} = 0.999 \pm 7.5\,e{-}4$, $n$ denotes the size of the data set, $d$ its dimension and $k$ refers to the number of clusters sought in the data set

| Data set $(n, d, k)$ | Algorithm | SSE | Time (sec) | $N_{entro}$ |
|---|---|---|---|---|
| S2 (5000, 2, 15) | RKM | $1.359\,e{+}13$ | 3.89 | 0.999565 |
| | BKM+ | $1.331\,e{+}13$ | 0.26 | 0.999737 |
| S4 (5000, 2, 15) | RKM | $1.594\,e{+}13$ | 6.76 | 0.998987 |
| | BKM+ | $1.577\,e{+}13$ | 0.16 | 0.998999 |
| vowel (871, 3, 6) | RKM | $3.175\,e{+}7$ | $4.68\,e{-}2$ | 0.999105 |
| | BKM+ | $3.169\,e{+}7$ | $0.71\,e{-}2$ | 0.999120 |
| user knowledge (403, 5, 4) | RKM | $7.052\,e{+}1$ | $8.50\,e{-}3$ | 0.999183 |
| | BKM+ | $7.039\,e{+}1$ | $2.04\,e{-}3$ | 0.999031 |
| ionosphere (351, 34, 2) | RKM | $2.424\,e{+}3$ | $4.14\,e{-}3$ | 0.999140 |
| | BKM+ | $2.432\,e{+}3$ | $2.67\,e{-}3$ | 0.999043 |

The results for the three real-world data sets user knowledge, vowel recognition and ionosphere are summarized in Figures 5, 6 and 7. The results are mostly consistent with that of the artificial data sets. When the focus is on the balance their results are almost identical. However, when the focus shifts towards optimizing the clustering quality, BKM+ produces better clusterings. Ionosphere is an exception where RKM provides better results in SSE for most part of the scale. Also, the speed advantage of BKM+ almost disappears with this data set due to its higher dimension.

Finally, the main results are summarized in Table 3 by fixing $N_{entro}$ to be about 0.999. At this point, BKM+ provides slightly lower SSE-values (0.63% on average) with a faster running time (77.46% on average).

### 6.3  Discussion

A problem that arises when RKM is used with SDCS or $N_{entro}$ is the difficulty to find an appropriate value for the balance parameter $\lambda$ since it is not possible to predict the resulting balance quality. The recommended range of values given by the authors (Lin et al., 2019) only fits for some data sets. Most data sets require significantly higher values for $\lambda$ and then the only way to find a suitable range is by trial-and-error. Besides, RKM tends to produce clusterings that are optimizable with respect to both the clustering and the balance quality if $\lambda$ is chosen too small.

In this sense, BKM+ is much easier to handle because the resulting balance degree does not depend on a parameter $\lambda$, but only on the point in time at which the algorithm stops increasing the penalty term. A clustering of a desired balance degree can easily be obtained by using the balance criterion as termination criterion.

The experimental results also allow to draw conclusions about the beneficial application of the proposed algorithm. First, the algorithm cannot compete with the other clustering algorithms in terms of the clustering quality when the focus is set on balance. It is not made for this because whenever the last clusters are resolved by shifting data points from larger to smaller clusters, there are always overlapping clusters. Therefore, it is not advisable to use this algorithm as a hard-balanced clustering algorithm even though this is possible.

However, the proposed algorithm has its strengths when it comes to soft-balanced clustering. Especially when the resulting clustering should match a certain balance requirement or the data set is not well-known in terms of its balance behaviour (in the sense that its balance in the case of an optimal clustering is unknown), this algorithm is much easier to handle than an algorithm using a balance parameter $\lambda$. Moreover, in the

soft-balanced case, it also has the shorter running time on most data sets. Therefore, it is advisable to use the proposed algorithm as a soft-balanced clustering algorithm, particularly when a specific balance criterion has to be achieved or the data set is rather unknown in terms of its balance behaviour.

## 7 Conclusion

We presented a balanced clustering algorithm based on the $k$-means algorithm which can be used for both soft-balanced and hard-balanced clustering. The main principle of the algorithm is an increasing penalty term, that is added to the assignment function of the standard $k$-means algorithm. The penalty term of a cluster is the larger the more data points a cluster contains. This way, smaller clusters are favoured when assigning the data points.

The main difference to similar methods following the approach of an additive bias in the assignment function of the standard $k$-means algorithm is the way the size of the penalty term is determined. While other algorithms use a constant factor by which the size of a cluster is multiplied to obtain the penalty term of the cluster (Althoff et al., 2011; Liu et al., 2018), the proposed method uses a factor that increases with each iteration. Further, this factor is not defined by an explicit function, but is computed anew in each iteration of the algorithm based on the current assignments and locations of the cluster centroids. We justified this additional computational effort by the individuality of each data set.

When used as a soft-balanced clustering algorithm, a characteristic of the proposed algorithm resulting from the increasing penalty term is the way in which the trade-off between the clustering and the balance quality is determined. Instead of using a rather non-intuitive parameter $\lambda$ like many other soft-balanced clustering algorithms do (Althoff et al., 2011; Li et al., 2018; Lin et al., 2019), the resulting balance degree only depends on the point of termination. Therefore, in contrast to other soft-balanced clustering algorithms, this approach enables an explicit specification of the trade-off between clustering and balance performance.

In the experimental results, we compared the proposed method to the regularized k-means algorithm (RKM) by Lin et al. (2019, a hard- and soft-balanced clustering algorithm) and the balanced k-means algorithm (BKM) by Malinen & Fränti (2014, a hard-balanced clustering algorithm). When testing the hard-balanced version of the proposed algorithm, both RKM and BKM are superior in terms of the clustering quality. However, regarding the running time, the proposed algorithm is comparable to RKM and several orders of magnitude faster than BKM. In the soft-balanced case, no algorithm is always superior to the other one. In general, the proposed algorithm returns the better clusterings if the focus is primarily on optimizing the clustering quality, and RKM returns the better clusterings if the focus is shifted towards the balance quality. In terms of the running time, the proposed algorithm takes less time on almost all data sets.

Moreover, the already mentioned advantage of the proposed algorithm that the trade-off between the clustering and the balance quality depends on the point of termination instead of a balance parameter $\lambda$ can also be seen in practice. When applying RKM, which is using this parameter to determine the resulting balance degree, there is always a risk of choosing $\lambda$ too small and obtaining a clustering that is still optimizable with respect to both the balance and the clustering quality. This situation does not occur when using the proposed algorithm, since the algorithm can always be continued to ensure that no following clustering is better than the returned one in terms of both the clustering and the balance performance.

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
