# OpenReview forum: "Balanced k-Means Revisited"
_TMLR — Rejected by TMLR_

### Review · Reviewer_DofD · 2023-01-30

**Summary Of Contributions:**

The paper proposes a new heuristic for balanced k-means clustering (denote BKM+).
It is a simple extension of existing ideas.  There are many methods in this subarea of data mining, and the paper does not significantly distinguish from existing ones or motivate the change.  Under a limited emperical comparison to only 2 existing methods (one written in MATLAB) it shows occasional improvement in some metrics, and often some runtime improvements.  It is not clear if the runtime improvement are due to novel approaches introduced in the paper, more careful implementation, or simply using C++ instead of MATLAB.
For an algorithm that balances the SSE criteria with a similar-size criteria there are unsurprisingly some parameters needed.  While some heuristic guidance on the parameters is provided, a full justification or exploration of the parameter space is not provided.

**Audience:**

No

**Claims And Evidence:**

No

**Requested Changes:**

Overall, I am not in favor of accepting this paper as I do not feel it demonstrates a meaningful advancement in the state of machine learning.  For acceptance I would expect at the minimum that the paper:

  1.  motivates the need for the claimed improvements in balance k-means clustering

      a.  While some motivations are mentioned on page 3, it is not clear why the existing solutions are not sufficient.  Will there be any tangible benefits for those application for the improvements claimed?

      b. at least one data set used should demand a solution to balanced k-means (the data sets used are either synthetic or "real" classification data sets which have labels so a classifier would be more appropriate.

  2.  if runtime is the motivating factor then it should provide a fair comparison that means:

      a. no MATLAB to C++ comparisons, all methods should be on equal footing

      b. the experiments should be demonstrated at scale where runtime is an issue (or explain why a  few seconds different is meaningful for some application)

      c. more than 2 competitors should be compared unless it is very clear those are the clear dominant state of the art.  For instance, at scale approximate methods should be considered as this method does not guarantee it reaches a global optimum anyways.

  3.  An ablation study on the choice of parameters of algorithms and data

      a.  Some plots show the effect of changing the algorithm parameters c and f.  The paper should justify experimentally why the default value make sense, and why the suggested parameter range is a good one

      b.  A synthetic study of what happens as datasets grow in parameters, n, d, and k.  As long as synthetic data is used, it is recommended to start with some detault values, and vary the synthetic data so to change n, d, k and show plots of how the algorithm performance varies in these parameters.


**Strengths And Weaknesses:**

Strengths:
  * experimental study has a lot of data sets

Weaknesses:
  * the improvements are poorly motivated
  * the runtime experiments are limited and compare C++ (theirs) to MATLAB (prior) in runtime
  * the parameter space of algorithms or data is not carefully explored
  * no guarantees on improvement or accuracy or runtime are provided about the algorithm

---

> ### Author Response · Authors · 2023-02-06
> **Comments to Reviewer DofD for Paper759**
>
> 1a.
> We thought TMLR papers are reviewed for validity instead of significance. But the numbers should be clear there are tangible benefits:
> - The improvement of BKM+ is huge over BKM (5000s vs. 0.18) on S1 (Table 2) in case of hard balance;
> - The improvement is remarkable over RKM (3.89s vs. 0.26s) in case of soft-balance.
> When we looked at other soft-balanced clustering algorithms, we did not find an algorithm where you could say, that you want a clustering of a data set that has a standard deviation in cluster sizes (SDCS) of at most 0.1% of the size of the data set. To get this result by another algorithm you have to try different values for non-intuitive balance parameters or set constraints for the cluster sizes (which is stricter than setting a constraint for the SDCS). Using BKM+ you can just set the termination criterion as a function that returns true, if the SDCS is less than 0.1% of the data size while using the default parameters c = 0.15 and f = f_iter.
>
> 1b.
> Datasets do not demand balance, applications demand. If we were to find real clusters, we would not need any balance criterion. If the application is to process the data in parallel then every data set demand balance. Unbalance is probably the most obvious example as normal (good) algorithms would divide it 2000, 2000, 2000, 5x100 clusters but best utilizing resources in parallel computing would require different (balanced) partition.
>
> Santa TSP dataset could be added for an example outside clustering (see: "Solving the large-scale TSP problem in 1 hour: Santa Claus Challenge 2020") but most likely there would be nothing much interesting to observe.
>
> Time complexity of algorithms like ("A fast minimum spanning tree algorithm based on K-means") assume balance clustering for the O(NlogN) time. Unbalanced data would be O(N^2). Normal k-means was applied in the but the problem noted. Our method would resolve the issue.
>
> ------------------------------------
> 2a.
> The only algorithm in Matlab is BKM, and the time difference is 5000s (BKM) vs. 0.18 (BKM+). While C++ implementation could improve it significantly, it would not solve the time complexity O(N^3). Thus, even C++ would be significantly slower. Even if it speeds up by factor 100, it would still be 50 against 0.18. This is why we did not consider it worth to re-implement this algorithm by C++. It would take time which the author responsible of the implementation has only weekends, and the Hungarian algorithm used in it is rather complex; hardly worth the trouble.
>
> 2b.
> About few seconds difference, it can make huge difference in practical applications if the first other one is 3.89s and the other 0.26s.
> - The first application remains interactive (<10s).
> - The second application would be real-time (<1s).
>
> The difference would also become more obvious if one scaled-up the data size from N=5000 to N=50,000 increasing the results to 38.9s and 2.60s. We already have Birch datasets (N=100,000) see Table 2. But generating plots for it would require roughly 1 month: each data point is average of 100 runs -> 100 data points x 100 runs -> 10,000 runs; 10,000 runs x 90 sec -> 900,000 sec = 15,000 min = 250 h, even if let it run for 8 hours every night it will take more than a month. And then we have only have one plot.
>
> 2c.
> We do not know what is "scale approximate method". An exact literature pointer would be needed here if something important. Currently we have added results for those methods working software been available.
>
> ------------------------------------
> 3a.
> Graphical plots of joint analysis of the parameters f and c already exist for S1, S3 and Wine but were left out from the paper. We can add them if needed. The results are not particularly critical on the choice of parameters.
>
> 3b.
> This has partly done already if one looks closer Table 2. The datasets A1, A2, A3 vary k (20, 35, 50) but also n (3000, 5250, 7500), and  datasets S1, S2, S3, S4 vary cluster overlap. These follow the setup from "K-means properties on six clustering benchmark datasets" where it was noted that dimensions do not play significant role but overlap does. The benchmark involves also other datasets (G2 and subsets of Birch2) that could be used but to see "what happens". BKM is too slow to do much more additional tests. Results with RKM is about equal in hard balance, and soft balance there might be something more to find out if we extended Table 3 to cover also A1-A3 or run Birch2 subsets but but all these easily takes lots of time with limited expected further insight.

---

> > ### Comment · Reviewer_DofD · 2023-02-18
> > **reply**
> >
> > Thank you for the careful reply.  I am less concerned about some of the points, or will be if much of this discussion is incorporated in the paper.   However, the lack of convergence that the other reviews raise is still concerning.
> > Let me follow up on some points.
> >
> > ## Re 1b.
> > | Datasets do not demand balance, applications demand.
> > Yes, this is what I meant to imply.  But no experiments are provided on datasets that are tied to applications that demand clustering balance.
> > And as mentioned, for the data sets used, if there are indeed sensical clusters, they may not have balance.  So why are these results useful?
> > Would the proposed heuristic show similar improvement on such data sets?  Would incorporating the cost of this algorithm vs. ones that do not attempt to control the balance lead to an improvement in the full task?
> > As it is, the paper seems to show results on data sets unconnected to applications where a balanced clustering would be useful.
> >
> > ## Re 2a.
> > Comparing runtime of C++ vs. matlab code is just not a good controlled experiment, and so does not provide very useful evidence in runtime.  I agree re-implementing the Hungarian algorithm in C++ is not useful, but there is code online.
> > A careful asymptotic analysis may be a suitable replacement.  But without a control on the number of iterations, this does not give a fair comparison either.  If the paper proposes running until it converges, or some upper bound number of iterations is performed (and this upper bound is justified), then this may be a suitable comparison.
> >
> > ## Re 2c.
> > This statement would have been more clear if phrased as "approximation methods should be considered at scale."
> > The point that this was trying to make, is that since the heuristic does not have any guarantees, then when the point is runtime and scalability, then approximation algorithms should be considered.  For instance, this paper using coresets:
> > https://papers.nips.cc/paper/2014/hash/861dc9bd7f4e7dd3cccd534d0ae2a2e9-Abstract.html
> > and/or follow-on work.
> >
> >
> > ## Re 3a.
> > Yes, I think those plots would be useful.  They would help justify the recommendations in the paper for those parameters.

---

> > > ### Author Response · Authors · 2023-02-28
> > > **DofD discussion**
> > >
> > > 2a.
> > > The reviewer agrees our claim that "Datasets do not demand balance, applications demand" but then claims that "no experiments are provided on datasets that are tied to applications that demand clustering balance". So any data has potential application demand. We considered adding the TSP data but uncertain if it possible to run further experimental results at this stage. The NIPS paper cited below shows perfectly that any data can serve as application.
> > >
> > > We have visual results of Unbalance dataset which we can add. They show how the balance constraint provides less meaningful clustering. This is inevitable when balance is forced even when data set is highly unbalanced.
> > >
> > > Why are these results useful? If applications demand, we MUST have balance no matter how ugly the clustering. We will add a bit more discussion on that.
> > >
> > > Would the proposed heuristic show similar improvement on such data sets? The results in Table 2 already contain Unbalance dataset for hard balance case. The improvements are the same (HUGE) compared to BKM, but insignificant compared to RKM. The improvements over RKM comes in the soft balance as RKM generalizes not so well to the soft case. We also note that originally RKM was not even defined for such case, it is just our own generalization. So basically we would not even have a competitor.
> > >
> > > 2c.
> > > The time complexity is O(N^3) vs. O(INk) where I is the number of iterations and varies from 15 to 800 (Birch1 with N=100,000, k=100). Even then the result would be 100,000^3 = 10^15 vs. 10^10. So theoretically we are 100,000 times faster. Implementation differences using C++ vs. matlab code is really insignificant. We can add this discussion in the paper if the reviewer otherwise cannot see the huge difference.
> > >
> > > About re-implementing Matlab code: it would be just waste of time, no matter how simple or easy. At this stage I can say that the author implementing the methods resigned from the project because of the overwhelming request in this review process, and at the moment, we are not able to do any new coding in any reasonable time.
> > >
> > > We earlier consider tau-balanced variant for which software was available but it was buggy and never worked. Others did not have source codes available. All the results were also ran more than 1 year ago and the computer used for testing is no longer even use. So adding any new time results would basically mean re-running them all. This is already such overwhelming request we have to deny.
> > >
> > > We agree it would be interesting to have wide comparison. But this paper is not a review paper. It was also turned down by JMLR in which such request might sound more reasonable. The current results are already enough for a proof-of-concept and we can add more of those already existing that was originally excluded from the paper to save space.
> > >
> > > A careful asymptotic analysis was suggested to be "a suitable replacement" but "without a control on the number of iterations it would be unfair". We will add the above discussion which should clearly show the time complexity differences to BKM and BKM+. We also remind the reviewer the fact that RKM is implemented by C++ and soft balance results are already fair. So we do not see any valid point for criticism here anyway.
> > >
> > > 3a.
> > > The reviewer clarified the comment "approximation methods should be considered at scale." and claimed that "since when the point is runtime and scalability, then approximation algorithms should be considered" and pointed to a paper: "Distributed Balanced Clustering via Mapping Coresets" by Bateni, Bhaskara, Lattanzi and Mirrokni, NIPS 2014. Its main contribution is:
> > >
> > >   "given a single machine alpha-approximation algorithm for a clustering problem (with or without balance constraints), we give a
> > >   distributed algorithm for the problem that has an O(alpha) approximation guarantee.
> > >
> > > Sorry but we disagree. That paper focus on distributed clustering and provides very little relevant information to our work:
> > >
> > > - It provides MapReduce framework for distributed clustering (not clustering algorithm itself)
> > > - There is no time complexity mentioned
> > > - There is no single running time included
> > > - It focus on using hundreds of processors but still not reporting how much time it takes
> > > - There is no pointer to software or even pseudo code
> > > - Its only claims space complexity which looks highly impractical O(N^2 * poly(k)).
> > >
> > > We find very little relevance to our work. Approximation algorithms should be considered when we need QUALITY GUARANTEE. But we seek for EFFICIENT solution both hard and soft-balanced criterion. The above paper consider only hard case where we can already manage N=100,000 with 1 minute by single processor.
> > >
> > > And yes, we will add the plots for the effect of parameter as they already exists.

---

> > > > ### Comment · Reviewer_DofD · 2023-02-28
> > > > **response**
> > > >
> > > > I am still not convinced about the demonstration of the importance of these results.  I will defer that to area chair to resolve.
> > > >
> > > > I also still don't think its good practice to publish runtime comparisons with C++ vs. matlab (unless it is something matlab is optimized for like pure matrix operations). I am convinced that the proposed algorithm will provide a result faster than the one with matlab code in practice, regardless of language.
> > > >
> > > > As for the approximation algorithms papers, that paper is the seminal one.  There are several follow-on works (which took me just a couple minutes to find):
> > > >  https://www.sciencedirect.com/science/article/abs/pii/S030439752030400X?fr=RR-1&ref=cra_js_challenge
> > > >  https://arxiv.org/abs/1704.02515
> > > >  https://arxiv.org/abs/1910.00788
> > > > as well as this more recent one (2023): https://link.springer.com/article/10.1007/s10878-022-00980-w
> > > > Admittedly they are theoretical papers, but deserve to be cited and discussed.  For instance, are their algorithms comparable.  They should not be dismissed as approximate algorithm, since the one your paper proposes is only heuristic.
> > > >
> > > > A few more minutes of digging finds this paper on "fair" clustering:
> > > >  https://arxiv.org/abs/1812.10854
> > > > that at least claims some relation between these two formulations in the k-means setting.  My preliminary interpretation is that balanced/constrained clustering can be a restricted special case of fair clustering.  And at least these widely-cited papers:
> > > >   NeurIPS 17: https://proceedings.neurips.cc/paper/2017/hash/978fce5bcc4eccc88ad48ce3914124a2-Abstract.html
> > > >   ICML 19: http://proceedings.mlr.press/v97/chen19d.html  (this formulation seems a bit different)
> > > > have experiments, so are not just experimental.  There looks like other work in this space that seems potentially related.
> > > >
> > > > Sorry to throw more references at the last minute, and perhaps there is a simple distinction.  But it seems like something the initial paper should have addressed.

---

> > > > > ### Author Response · Authors · 2023-03-09
> > > > > **Further discussion with DofD**
> > > > >
> > > > > 1.
> > > > > First, we cannot estimate the importance of the results but the topic is clearly relevant (our original BKM paper was our most downloaded for years despite of being so impractical). We rather let it to the readers to decide. The method works, it is not perfect, but provides good trade-off between clustreing quality and balance. We merely ask to evaluate its validity and let the readers decide. All our data and software is publicly available for anyone want to use them. This is not the case of most existing solutions.
> > > > >
> > > > > And while we cannot extend the comparison to other methods, this is what we know:
> > > > > - BKM is impractically slow
> > > > > - RKM was designed for balanced-constrained case only; our tuning to balanced-driven case is ok, but inferior
> > > > > - We also tested third software available but it never produced valid results
> > > > > - All others considered would require own implementation.
> > > > > - Their parameter setup is non-intuitive and expected to cause more troubles to find suitable cobination than our method.
> > > > > - They all variant of sub-optimal k-means.
> > > > > - Likely further improvements would come from replacing k-means by swap-based or another better optimizer, not by better cost function.
> > > > >
> > > > > If we were writing review paper then it would be different story. Now this was just a result of one student thesis which itself took long time and lots of efforts, but this work is no longer continued. Putting new major resources to this is simply not possible.
> > > > >
> > > > > 2.
> > > > > We disagree on publishing Matlab run times. Even though direct comparison of EXACT difference cannot be made, it still gives clear picture on the ORDER OF MAGNITUDE difference. There is nothing misleading in this if we CLEARLY MENTION in text that some part of this difference comes from using Matlab instead of C++. Better to show the numbers than hide, or do you really prefer NOT to add them? Re-implementing that method is simply unattractive and unrealistic option at this stage.
> > > > >
> > > > > The following text addition will be made at the end of Section 4.5:
> > > > >
> > > > > Nevertheless, the algorithm is several orders of magnitude faster than BKM, which requires O(N3) just for the assignment step whereas BKM+ requires O(INk) for the entire algorithm. Here I is the number of iterations which varies from I=15 to 800 with our datasets. The longest iterated is Birch1 having parameters I=800, N=100,000 and k=100. This corresponds to 8×109 steps whereas the BKM assignment step alone would require 100,0003 = 1015 steps. The difference is huge.
> > > > >
> > > > > 3.
> > > > > What it comes to the approximation algorithms, no matter how seminal paper that is, there are two completely different objectives:
> > > > > - Ours: Balanced-driven algorithms aiming at good quality clustering EFFICIENTLY.
> > > > > - Approximatiopn: To obtain result that has UPPER BOUND
> > > > >
> > > > > The second cases are usually algorithms that are either extremely impractical (very slow) or the upper bound is so high that they are practically meaningless as other heuristics always provide much better result even if no upper bound given. Having upper bound has theoretical interest usually but very little practical usefulness. Whether these results apply to balance-constrained variants or not, is really irrelevant unless the algorithm itself is particularly good AND efficient. The first pointed paper did not even have any run times (only relative improvements).
> > > > >
> > > > > In general, we do appreciate literature pointers even if not directly relevant as one can never be sure if not missed something relevant. Especially when given exact paper title or direct web pointer instead of vague hint as some reviewers tend to do. But these are rather late stage with "couple of minutes search" which is rather un-convinsing. Anyone can find any seemingly related papers fast but finding something really relevant takes time and can be like seeking needle in the hay stack. We still appreciate new pointers although not going to have much time to study those unless direct relevance.
> > > > >
> > > > > About fair clustering: this is somewhat interesting as they indeed share some relation to balance clustering. We also think balanced-clustering might be a special case of fair clustering, which seems more challenging problem. But this is all speculation and out of scope here. Fair clustering just adds so many new constraings that it is questionable is it even a clustering problem anymoire as the goal is just to split data into groups in which same features should be equally distributed. Clustering requires also similarity of the items, not just fairness. It might better modelled as some other graph or assingment problem. Thanks for pointhig this out. So far we have not found application for fair clustering and have therefore not studied it deeper.
> > > > >
> > > > > We are now going to implement the changes to the manuscript that we raised up by all the reviewers and submit revised version hopefully sometime next week. If something important missing, please address us the remaining issues we might have overlooked.

---

### Review · Reviewer_UTs9 · 2023-02-08

**Summary Of Contributions:**

This paper proposes an algorithm for the soft-balanced k-means problem. The proposed algorithm is called balanced k-means revisited BKM+, which works as a post hoc local search strategy. First, the algorithm runs k-means, and then it successively changes each point's partition to reduce the cluster imbalance.

The results show that the proposed algorithm is very competitive with the regularized k-means algorithm RKM. The main advantage of the proposed algorithm is that it is more intuitive than the baseline because it has a termination strategy that specifies the tolerance on the cluster balance, such as (max cluster size - min cluster size), rather than a regularization parameter that trades off cluster quality with class balance as in the RKM.


**Audience:**

Yes

**Broader Impact Concerns:**

I have no concerns about the ethical implications of this work. Its main contributions is a variant of a well-known clustering algorithm.

**Claims And Evidence:**

No

**Requested Changes:**

My most crucial point is the answer to my first question:

How do we know this algorithm will converge to a local optimum? Without a solid answer to this question, it is hard to argue that this is an improvement or a contribution to previous work.

The authors should also consider a major review to improve the clarity of the submission; a simple and concise description of the algorithm and why this algorithm works would make this paper much stronger. More details were made in the points to improve above.


**Strengths And Weaknesses:**

**Strengths**

Some individuals in the TMLR's audience would consider this an important topic. A balanced k-means solution is relevant for academics and practitioners, so a simple yet effective solution for this problem interests this community. A simpler algorithm that lets you better control the cluster balance is also a nice feature of the proposed approach.

**Opportunities for improvement**

*Analysis*. How do we know that this algorithm will converge to a local optimum? It seems that in section 3.2, there is an effort to always decrease the objective function, but Equation 7 in Section 3.3 just ignores that design choice. This paper would be much stronger if the authors argued that this algorithm makes progress in optimizing the proposed objective function.

*Presentation*. This paper could be more precise in its definitions. The first paragraph of Section 3 needs to be more precise; the reader must make a lot of assumptions in their mind to correctly read Equation 2. As written, the sum of the penalties could be considered a constant value.

The following claim seems misleading to me.

> In this way, the desired balance degree can be specified precisely, and the algorithm can always be continued to ensure that it does not produce a better clustering with respect to the SSE satisfying the given balance requirement in future iterations.

No optimality guarantees exist for any given run of the proposed algorithm, or at least none was given. And any statement about the best SSE seems misleading since K-means itself is a hard problem.

In the "results" section, consider reporting the standard deviation of the results. The column with 'best' should be suppressed since the algorithm is nonconvex or replaced by the median. I would like to know if reporting the root-mean-square error in a similar fashion that you did the SSE would result in a better metric value for reference and comparison.

Throughout the text, the presentation could be more concise and objective. For example, sentences starting with "apparently" and "probably" could be rephrased.

After reading the introduction, I thought this algorithm had no complex parameters that needed to be tuned, but the algorithm does have parameters c and f that could be adjusted. I understand now that the authors were talking about the termination criteria vs. the trade-off parameter on the baseline. Still, a sensitivity analysis of these parameters would increase the evidence that this algorithm is easier to use than previous work.

As a minor comment, the authors could be more consistent with the message and audience. For example, in the sentence below:

> Clustering denotes the unsupervised classification of objects into groups, called clusters.

I think it's a little bit hard to believe that someone that understands what "unsupervised classification" means does not know what cluster is. This is an anecdotal example, but there are a couple of sentences, especially in Section 3, that the message could be streamlined if you considered what you want to communicate.

---

> ### Author Response · Authors · 2023-02-14
> **Comments to Reviewer UTs9 for Paper759**
>
> Analysis:
>
> We chose the objective function as SSE + \sum_{j=1}^{k} penalty(j) =  SSE + \sum_{j=1}^{k} scale(iter) * n_j. Therefore, the objective function depends on the iteration and changes in each iteration based on the value of scale(iter). The more iterations are done, the more the objective function weights the balance against the SSE (by design scale(iter) never becomes smaller) until finally the desired balance degree is reached. We can argue that in each iteration the respective objective function is locally optimized for the given balance, but the balance keeps increasing.
>
> Maybe the missing global objective function is the main difference to other algorithms. Other algorithms require a balance parameter set beforehand, which determines the objective function. When the algorithm is applied, we see what the balance of the resulting clustering is only at the end. So, the balance is the "surprise". The proposed algorithm work opposite. It has a balance constraint as a termination criterion. We then apply the algorithm for the given balance criterion, but we do not know how the final objective function will look like. So for us, the last objective function is the "surprise".
>
> Presentation:
>
> We can try to improve that by introducing the assignment step first, which might make it easier to digest.
> Also other comments about readability we can revise paying attention to the mentioned issues aiming at better objectivity.
>
> Misleading claim:
>
> Yes, that is incorrectly formulated. The original idea of this sentence was the observation that sometimes it can be beneficial to continue the algorithm with the last penalty term fixed as soon as the desired balance degree is reached. It is explained in more detail in the second paragraph of 4.4, but also there it is misleading. We can fix these.
>
> Column best + st.dev:
>
> We can replace best by standard deviation. It seems to make more sense. Wheter SSE or Root-MSE would be better, we do not know. We have selected SSE for consistency without our previous research.
>
> Parameters:
>
> Graphical plots of joint analysis of the parameters f and c already exist for S1, S3 and Wine but were left out from the paper. We can add them if needed. The results are not particularly critical on the choice of parameters.
>
> Convergence:
>
> There is a proof that the algorithm converges after a finite number of iterations with the desired balance if f > 1but the proof is not currently included in the paper. A rough sketch of the proof is below: (a) If scale(iter) is larger than the maximal squared distance between two data points of the data set denoted by d_max, the penalty part of the assignment function always outweighs the squared distance to the centroid part. Hence, after only one iteration using this scale(iter)-value, a balanced clustering is obtained and a balanced clustering always satisfies the desired balance. (b) So it is sufficient to show that there exists a value for iter, such that scale(iter) > d_max. scale(iter) = f * scale_min(iter - 1) > f * scale(iter -1) = f * f * scale_min(iter - 2) > ... > f^(iter-1) * scale_min(1)  (w.l.o.g scale_min(1) > 0 ow. the data set is already balanced). To show: There exists a value for iter such that f^(iter-1) * scale_min(1) > d_max <=> f^(iter-1) > d_max / scale_min(1) This is obvious for d_max / scale_min(1) <= 1 (because f > 1), o.w. take for example the natural logarithm, this results in (iter-1) > ln(d_max / scale_min) / ln(f).

---

> > ### Comment · Reviewer_UTs9 · 2023-02-21
> > **Thanks for the reply!**
> >
> > Thanks for the comments! I think my read on this paper was very similar to the review above, and the other reviewer also brought up some crucial points about the experiment setup. So I only have a little to add.
> >
> > The one thing that I want to stress is the improvement in the presentation. The paper needs to be clear, and the language should be precise and straightforward. I also want to agree with the "Replies and further questions/comments" of reviewer 3onZ, specifically with the first bullet point.
> >
> > If I recall correctly, open review lets you upload a new version of the paper, so it would be ideal to have the new version available, organized to address all the comments, before this Friday.

---

> > > ### Author Response · Authors · 2023-02-28
> > > **UtS9 discussion**
> > >
> > > Sorry we could not find time for this yet, but we have now listed the required updates based on the discussion and will work on the updates. We still need to digest the three last bullets on 3onZ and then work on this.
> > >
> > > Currently we are able to:
> > > - Improve writing (with professional language checking)
> > > - Clarify most essential points raised up
> > > - Add already existing material about parameter sensitivity, demonstrate the effect when data is highly unbalance but balance is requred
> > >
> > > We are unfortunately unable to:
> > > - Implementing any new methods
> > > - Also highly possible not able to run more experiments.
> > >
> > > We are now working on those. If able to complete by the end of this week we will submit. After that there will be break by both authors in this project. Implementing something more would require recruiting a new person which at this stage is highly uncertain.

---

### Review · Reviewer_3onZ · 2023-02-10

**Summary Of Contributions:**

The paper proposes a simple modification to existing balanced k-means algorithms in which the so-called "bias" added to the SSE objective is updated with each iteration, and its scale is updated in a manner which is not only data dependent but dependent on the current "state" of the algorithm. Experiments indicate the proposed approach has potential relevance, especially in the soft-balance scenario, by its comparative performance with regularised k-means (RKM).




**Audience:**

Yes

**Claims And Evidence:**

No

**Requested Changes:**

Major:
- I would like to see the justification for setting scale(iter) better rationalised. The basic idea behind it is sensible, but the assumption on which the calculation is based seems dubious. Is there a way to improve this, or perhaps it can be guaranteed that it will actually work? That is, can you guarantee that the setting scale(iter) will definitely lead to at least one point moving even if points are moved "asynchronously"?
- I think the experimental results need to include comparisons with more alternatives, to gain a better sense of context.

Minor:
- Some of the statements are not clear to me as a reader. Can you improve clarity?
-- Repeatedly it is mentioned that one will get overlapping clusters if the scale is increased too quickly. Is this not technically always possible, no matter how small f > 1? Whether or not the clusters overlap will also, if I understand, depend on the order in which points are processed in a given iteration.
-- In the discussion at the end of the experiments, the sentence "Besides, RKM tends to produce clusterings that are optimizable with respect to both the clustering and the balance quality if λ is chosen too small. In this sense, BKM+ is much easier" confuses me. Can you clarify what you mean?
-- Something is wrong syntactically in the sentence "The penalty term of a cluster is the larger the more data points a cluster contains."
-- Referring to the two objectives of minimising SSE and optimising balance as "contradictory" is, in my opinion, a poor choice of words. I suggest choosing another word since sometimes they may correlate, but will not always do so.
-- The notational change from scale(iter) to p_{now} seems unnecessary. Am I missing something?

**Strengths And Weaknesses:**

Strengths:
- The idea of incrementally increasing the penalty term is appealing and intuitive and strongly reminiscent of "continuation" schemes in optimisation, where an initial relaxation of the problem is considered and the objective is iteratively modified to approach the "actual" objective for the main problem.
- The idea behind the data and algorithm-state dependent increments in the penalty is appealing.
- The method is clearly computationally viable for even relatively large problems.
- The guarantee of achieving a desired balance level (minimum cluster size), when compared with methods which do not have this guarantee, is a huge plus.

Weaknesses:
- The mathematical justification for the penalty scale term "assuming that there are no changes in the assignments and locations of the clusters j-old and j-new until the assignment phase of x_i in the next iteration." is crucial for their argument, but does not hold actually hold. Indeed, as far as I can tell there is no guarantee that the method will actually converge and supply a solution with the desired balance since in each iteration numerous reassignments can occur and therefore the "assumption" above would therefore be heavily violated.
- The authors boast that their algorithm is free of tuning parameters, but then suddenly "f" and "c" pop up and are just that. The addition of "f" may be to do away with the fact that their "assumption" from the previous point does not hold, but there seems not to be a practically sensible value for "f" which would guarantee convergence. The justification for using the "remaining fraction" parameter is not clear to me. Furthermore, if it is supposed to represent that a point hasn't fully left a cluster to
- As far as I can tell if p_{now} is set to zero initially, as described in the algorithm, then the algorithm will not actually work. If the initial solution is a k-means solution, and p_{now} = 0, then surely j+ is always equal to j-, and nothing happens. Can you correct my misunderstanding?
- The experimental section is a bit weak, as it only compares with one other computationally viable alternative, whereas the literature review suggests the existence of many more.
- As far as I can tell the output of the algorithm depends on the order in which the data are processed in each iteration. This is not a desirable property for an algorithm to have unless absolutely necessary.

---

> ### Author Response · Authors · 2023-02-14
> **Comments to Reviewer UTs9 for Paper759**
>
> Weaknesses are discussed below.
>
> Mathematical justification:
>
> Yes, it is true that the quoted assumption does not hold. The algorithm could be slightly modified to make the assumption true:
> (1) Perform one iteration of the algorithm without any assignments, only to compute the value of scale_min(iter) and remember the data point that leads to this value.
> (2) Assign the remembered point.
> (3) Update the affected cluster centroids.
> The problem with this alternative algorithm would be that even if it strictly follows the mathematical approach, is that in one iteration only one data point would change its cluster. This makes the algorithm very slow and impractical. So, while we have taken the mathematical consideration as an idea, we do not follow it strictly to make the algorithm usable in practice.
>
> There is a proof that the algorithm converges after a finite number of iterations with the desired balance if f > 1but the proof is not currently included in the paper. A rough sketch of the proof is below:
> (a) If scale(iter) is larger than the maximal squared distance between two data points of the data set denoted by d_max, the penalty part of the assignment function always outweighs the squared distance to the centroid part. Hence, after only one iteration using this scale(iter)-value, a balanced clustering is obtained and a balanced clustering always satisfies the desired balance.
> (b) So it is sufficient to show that there exists a value for iter, such that scale(iter) > d_max. scale(iter) = f * scale_min(iter - 1) > f * scale(iter -1) = f * f * scale_min(iter - 2) > ... > f^(iter-1) * scale_min(1)  (w.l.o.g scale_min(1) > 0 ow. the data set is already balanced).
> To show: There exists a value for iter such that f^(iter-1) * scale_min(1) > d_max
> <=> f^(iter-1) > d_max / scale_min(1)
> This is obvious for d_max / scale_min(1) <= 1 (because f > 1), o.w. take for example the natural logarithm, this results in (iter-1) > ln(d_max / scale_min) / ln(f).
>
> Free of parameter tuning:
>
> The main motivation for introducing f is to make the mathematical idea practical. If only one data point changes its cluster during the most iterations, the algorithm becomes very slow. Further, if f > 1, convergence can be guaranteed.
>
> The motivation for c is a bit more complicated. We did a detailed analysis of different ways of updating the cluster sizes because it was hard to find a suitable way. The analysis is not currently included in the paper because the update of the cluster sizes is not the main topic, it is only a tool to make the rest work.
>
> The proposed algorithm has no tuning parameters that influence the resulting balance. Both parameters c and f can be tuned, but they do not need to, because the algorithm works well with the default values. There also exists a detailed analysis of the impact of c and f which is not currently included in the paper.
>
> Misunderstanding
>
> If the solution is not balanced, p_next,i (line 11 of the pseudo code algorithm) will not be zero. There will always be a cluster that has too few data points, assume this is cluster j*, and a cluster containing too many data points, j-. If a data point x_i from cluster j- is considered, then PenaltyNext(i, j-, j*) will not return zero because the distance between x_i and the centroid of j* is larger than the distance to the centroid of j-, otherwise the initial solution is no k-means solution.
>
> Lack of comparisons:
>
> We have compared to those alternatives software have been found (and was working). Comparing with more algorithms is question of being able to select the most relevant, re-implementing and testing them. It is likely to take lots of time.  Currently we have focused on the proposed approach instead of full-scale survey of all approaches.
>
> Order of processing:
>
> Yes, that is true. This follows the on-line (sequential) variant of k-means where centroids are updated after every step. It is usually avoided when possible, but used when the standard batch variant of k-means would not converge. The balance criterion, while simple, prevents running classical k-means.
>
> For the experiments the data points of the data sets were shuffled to account for this. We made experiments to order the data points based on different properties, for example their distance to their centroids, but the results did not become better, only the running time increased especially for large data sets. Therefore, the ordering-approach was discarded.
>
> Requested changes:
>
> The idea behind the scale(iter) is that at least one point will be moved in every iteration. It guarantees that the algorithm will work and not going idle without progress. We can consider adding more algorithms in the comparisons but this would take quite some time as most of them have non-intuitive parameter settings and it might hard to get sensible results out of them quickly. All minor comments we can easily fix.

---

> > ### Comment · Reviewer_3onZ · 2023-02-17
> > **Replies and further questions/comments**
> >
> > - I appreciate that you could guarantee the assumption holds by first passing over the data as you describe, before moving a single point in each iteration, but that this would not be a computationally viable approach. However, using as the basis for a derivation an assumption which is known not to hold, and the violations of which may be very substantial, is not persuasive. I believe that to make the justification acceptable a discussion around the extent to which the assumption is violated would be necessary. Furthermore, since the algorithm is relatively simple, it seems likely there is a way to derive a value for scale(iter) which is guaranteed to move at least one point at each iteration based on the differences between points and their old and new cluster centroids. Perhaps I am wrong, but I unfortunately find the rationale in its current form to be insufficient.
> > - Regarding your proof. I should have been more precise. What I would like to see is some guarantee that convergence to a non-degenerate solution would occur. The fact that eventually scale(iter) ensures that the distance values do not matter means that your algorithm will converge but the solution could be meaningless from a clustering point of view.
> > - Regarding my misunderstanding: At the start of the algorithm you set p_{now} to 0. At iteration 1 you then set j+ to argmin_j ||xi - cj|| + p_{now} nj. But since p_{now} is zero, all assignments are based on the distances to centroids. Since you initialised with kmeans, all points are already allocated to their nearest centroid, hence j+ = j-. p_{next,i} is then infinite since nj- = nj+. This will be true for all points, and so it seems no updating ever occurs. Where am I going wrong?
> >
> >
> > I appreciate your argument for not including further comparisons in the experiments. The simplicity of many algorithms based on kmeans may be such that implementing them would not be time consuming. I am genuinely not sure how complex the methods you have cited are. In short, I accept your justification, but including more comparisons can only serve to improve the work. My concern is that to be interesting to the ML community I believe the paper would need either a stronger mathematical justification or stronger evidence that it is competitive with existing methods.

---

> > > ### Author Response · Authors · 2023-03-09
> > > **Discussion with 3onZ**
> > >
> > > 1.
> > > Not sure how to address this issue but the algorithm works and as planned. It can happen that in one iteration there is no point moving but after such an iteration we are in the case that the data did not change during one complete iteration and we can be sure that there will be at least one point moving during the next iteration. So, we can say, in every second iteration there is at least one point moving. So there is no complications or fault in this part as far as we know.
> > >
> > > 2.
> > > This point is more interesting and raised some concerned among ourselves as well.
> > > (a) First, such guarantee does not exist. The algorithm should not be driven so far that distances would not matter at all. Otherwise, it is possible to get a perfectly balanced clustering solution where some clusters are less meaningful.
> > > (b) Second, the previous iteratations have already created meanigful (less balanced) clustering and prevents complete degeneration. At the very last stages points will start to make sub-optimal and even strange moves if (close to) balance is forced. We do not expect other algorithms can handle such cases much better though.
> > > (c) Third, this is mainly the weakness of k-means that leads to sub-optimal solution. In case of balanced cost term, the effect might be more radical but eventually better optimizer would be needed to handle this. K-means worked surprisingly well for us so we did not seek better algorithms.
> > >
> > > We will add visual example using Unbalance dataset in the paper to demonstrate the situation and the following text (draft so far) to the discussion:
> > >
> > > The algorithm provides good trade-off between cluster size balance and clustering quality. However, if the algorithm is iterated so far that the distances would not matter anymore. Otherwise, it is possible to get a perfectly balanced clustering solution but having some less meaningful clusters. The assignments from the previous iterations will keep the solution mostly meaningful but the case with Unbalance dataset shows that the result is not perfect if iterated too long. Other k-means based balanced-constrained clustering are likely to have the same problem. In this regard, there are still room for further improvement. Swap-based algorithms have been previously used to overcome such problems of k-means. For example, problems of classical k-means mostly disappeared by using random swap (Fränti, 2018), clustering graphs by M-algorithm (Sieranoja&Fränti 2022) and clustering sets by k-swaps algorithm (Rezaei&Fränti 2023). The same approach might be worth considering for the balanced k-means as well. This is a point of future research.
> > >
> > > 3.
> > > Not sure how to clairfy this (help in this regard welcome) but it is true, that in the described case j+ = j-. But the computation of p_{next,i} does not depend on j+. The assignment of x_i to j+ is already possible with the current penalty, but the cluster that leads to the p_{next,i} value is a cluster to which x_i cannot be assigned in this iteration because the current penalty is a little bit too small.
> > >
> > > We hope these clarifications and planned revisions will clear the situation.

---

> > > > ### Comment · Reviewer_3onZ · 2023-03-12
> > > > **Final replies**
> > > >
> > > > Thanks to the authors for their response and clarifications.
> > > > 1. I see that if no point moves in an iteration, then it is as in the "slow" algorithm where you move exactly one point at a time. Except that now the increasing scale due to factor f is applied. I think it remains somewhat inelegant, but at least there is a guarantee to move at least one point every two iterations. It still seems, however, as though a more thorough treatment of this step could have been done in order to make this part of the paper (and algorithm) more persuasive.
> > > > 2. The sentence "We do not expect other algorithms can handle such cases much better though." is a bit presumptuous. I would have liked more explanation for why. In addition, I don't understand why the assertion "... this is mainly the weakness of k-means that leads to sub-optimal solution." is true. The discussion is around the fact that the only way to guarantee the approach proposed in the paper converges involves the possibility of converging to degenerate solutions which have no meaning in the context of clustering. Unfortunately the authors left their response to my concerns until almost the last minute and there is no remaining time to discuss these points.
> > > > 3. Thanks for the clarification, I had misread the "penalty next" step and interpreted it as it is described in the text in relation to Eq.'s (3), where the motivation is described.

---

### Decision · Action_Editors · 2023-03-16

**Recommendation:** Reject

**Comment:**

Overall, all three reviewers agree that this paper is not ready for publication.  One of the biggest issues is that, while the authors responded to some of the criticisms, the paper was never updated and so the reviewers could not look at the updates.  The reviewers continued to have several concerns with the paper, as noted in the reviews (correctness of the algorithm, presentation, and experimental results, to name a few).  It seems that this paper needs some major work and then a re-submission.

**Audience:**

Yes, there are people in the community that would be interested in this work.

**Claims And Evidence:**

Several of the authors noted deficiencies in the experimental results (one other baseline, and questionable comparisons).  There are issues with the analysis of the algorithm (e.g., no convergence guarantees).  There are clarity issues throughout the paper.

---

> ### Author Response · Authors · 2023-03-19
> **What? Why the journal does not give enough time for revisions**
>
> We have already stated in the discussions the planned actions, and kept collecting list of planned revisions and have been implementing those. We have also informed planned actions in the discussions. Maybe our schedule has been too optimistic and taken a bit longer than desired. But we have implemented them and planned to finalize the revision by Tue 21-March.
>
> In specific, we have addressed all comments about validity and clarity of the method, and did not observe further concerns. The algorithm converges and we did not see any doubts in the dicsussion anymore. There is no questionable comparison, and the huge time complexity difference should be obvious and will be underlined in the paper.
>
> The only requirement we disagree was the last minute requirement to add more references about approximation algorithms or fair clustering which have very little relevance to the paper.
>
> The revised version will be submitted Tue or Wed latest.
>
> Why such premature rejection not giving enough time?

---

> > ### Comment · Action_Editors · 2023-03-20
> > **Re: Revisions**
> >
> > The TMLR process is designed in part to expedite the decision process to provide decisions in a timely manner.  Given that all reviewers were still advocating for reject at the end of the discussion phase, with various remaining concerns, it's unlikely that the paper would have been ultimately accepted even with the changes implemented.  I would encourage the authors to resubmit as a new submission, once all revisions have been carried out.

---

> > > ### Author Response · Authors · 2023-04-05
> > > **Complete the process - deanonymize the reviewer**
> > >
> > > The editor suggested:
> > >
> > > - It's unlikely that the paper would have been ultimately accepted even with the changes implemented.
> > > - I would encourage the authors to resubmit as a new submission, once all revisions have been carried out.
> > >
> > > These two comments contradict each other. The editor speculates that all the reviewers were to reject the paper anyway even if revised, and still asks to re-submit. Why? The revisions were mostly done already but not given enough time to submit the revision. Re-submitting would also mean keeping the original reviewers names anonymous. We are expecting open review.
> > >
> > > We thank for the opportunity but decline the possibility to re-submit under such conditions.
> > >
> > > We already addressed all reasonable comments, and gave arguments why the rest are not done. There were some good review comments that helped to clarify the writing and adding more results on the critical situations. Especially the de-generation issue was useful.
> > >
> > > The one editor picked up:
> > >
> > > 1. Experiments: We tested three algorithms (C++, Matlab, broken) of which two working ones are reported. Having one obviously slow (BKM) implemented only in Matlab does not make the comparisons unfair. Just check the coments carefully considering its time complexity and the actual numbers. This is addressed in the revised version so clearly that should be impossible to miss anymore. Re-implementing BKM would be just a waste of time, un-interesting, and not going to provide anything significant. Other algorithms would likely require complete implementation and potentially excessive parameter tuning without going to affect the validity of our method in anyway. The significance is already explained as the fundamentally different way to select parameters. This is also clarified in the revised version.
> > >
> > > 2. Converge: the algorithm converges and we did not spot any issues. This is discussed in the revised version.
> > >
> > > 3. Clarity issues: We sipmlified intro, added results for the effect of parameters (not sensitive), time complexity discussion of BKM, additional motivation for the needf for balance (which should be quite obvious though), added visual results, added example for de-generation and discussion about its reasons. Language is already reasonable for understanding and text we did not spot anything really unclear.
> > >
> > > The only remaining issues were to consider whether the de-generation issue was worth to revise the method by post-processing, or just leave it as deficiency which happens in the hard balance case. And final proof-readiong of language.
> > >
> > > The TMLR process itself is unclear and it turns out revisions were expected already during the discussion, which was not clearly stated. Possible hard deadlines were also missed (if any) as the comunication become rather over-whelming.
> > >
> > > If we were working for this full-time, fine, but the student implementing the methods and making the experiments and most writing graduated already long ago and moved on, and could spend at most weekends for this (if even those). Eventually retired from the project due to overwhelming number of requests with comment "there are more important things in life". The guiding professor has only 24 hours a day, of which sleep, family, and dozens of other students require their share of time. This also happened in the middle (temporarily) re-locating to another country. The retirement of the student also removed the only LaTex knowledge (choice of the journal), and required recuiting third author to continue the process. In this light, comments like "authors submitted comments in last minute" (reviewer) and "timely manner" (editor), while harmless, are questionable as the end result is just further burdening for the authors.
> > >
> > > The original version was submitted to JMLR and recommended to transfer to TMLR (without review) based on selectivity with speculations "it will be likely rejected by reviewers". TMLR was marketed focusing more on validity than significance. But the end result is just the same: "reviewers are likely to reject it anyway". The method is valid and its results are as described in the paper, they are useful and relevant. The end result the paper is now just much longer, the process lots of time, and the remaining requests does not relate to validity as long as we could see.
> > >
> > > We thank the attempt to make open review journal with interactive discussion with the reviewers. We also thank those useful comments which are these days rare in many journals. We just cannot afford to spend more time on this as this one became another burden.
> > >
> > > Please complete the proces and de-anonymize the reviewers.

---

> > > > ### Comment · Action_Editors · 2023-04-07
> > > > **Re: complete the process**
> > > >
> > > > To clarify, I do not mean re-submitting with the original reviewers.  I mean submitting separately (perhaps to a new venue) after making the changes suggested by the reviewers.  Given the current opinions of the reviewers (all reject or leaning reject) I did not see a path forward here.  Given the relatively short timeline at TMLR, we had to make a decision even before all edits were made.